# Artemisinin-resistant K13 mutations rewire *Plasmodium falciparum's* intra-erythrocytic metabolic program to enhance survival

Sachel Mok [1], Barbara H. Stokes [1], Nina F. Gnädig[1], Leila S. Ross[1], Tomas Yeo [1], Chanaki Amaratunga[2], Erik Allman[3], Lev Solyakov[4], Andrew R. Bottrill[4], Jaishree Tripathi[5], Rick M. Fairhurst[2,9], Manuel Llinás [3,6], Zbynek Bozdech [5,10], Andrew B. Tobin [7,10] & David A. Fidock [1,8 ✉]

The emergence and spread of artemisinin resistance, driven by mutations in *Plasmodium falciparum* K13, has compromised antimalarial efficacy and threatens the global malaria elimination campaign. By applying systems-based quantitative transcriptomics, proteomics, and metabolomics to a panel of isogenic K13 mutant or wild-type *P. falciparum* lines, we provide evidence that K13 mutations alter multiple aspects of the parasite's intra-erythrocytic developmental program. These changes impact cell-cycle periodicity, the unfolded protein response, protein degradation, vesicular trafficking, and mitochondrial metabolism. K13-mediated artemisinin resistance in the Cambodian Cam3.II line was reversed by atovaquone, a mitochondrial electron transport chain inhibitor. These results suggest that mitochondrial processes including damage sensing and anti-oxidant properties might augment the ability of mutant K13 to protect *P. falciparum* against artemisinin action by helping these parasites undergo temporary quiescence and accelerated growth recovery post drug elimination.

[1] Department of Microbiology & Immunology, Columbia University Irving Medical Center, New York, NY, USA. [2] Laboratory of Malaria and Vector Research, National Institute of Allergy and Infectious Diseases, National Institutes of Health, Bethesda, MD, USA. [3] Department of Biochemistry & Molecular Biology, Huck Center for Malaria Research, Pennsylvania State University, University Park, PA, USA. [4] Protein Nucleic Acid Laboratory, University of Leicester, Leicester, UK. [5] School of Biological Sciences, Nanyang Technological University, Singapore, Singapore. [6] Department of Chemistry, Huck Center for Malaria Research, Pennsylvania State University, University Park, PA, USA. [7] The Centre for Translational Pharmacology, Institute of Molecular, Cell and Systems Biology, College of Medical, Veterinary and Life Sciences, University of Glasgow, Glasgow, UK. [8] Division of Infectious Diseases, Department of Medicine, Columbia University Irving Medical Center, New York, NY, USA. [9] Present address: Astra Zeneca, Gaithersburg, MD 20878, USA. [10] These authors contributed equally: Zbynek Bozdech, Andrew B. Tobin. ✉ email: df2260@cumc.columbia.edu

*P*lasmodium falciparum parasites, the primary etiological agent of severe malaria, caused an estimated 229 million clinical cases and 409,000 deaths in 2019, predominantly in young African children[1]. Disease is caused by the asexual blood stages, which complete their intra-erythrocytic developmental cycle (IDC) every ~48 h. Globally, first-line treatment depends on artemisinin (ART)-based combination therapies (ACTs), which benefit from the exceptional potency of ART derivatives. These derivatives are activated in the parasite by hemoglobin-derived $Fe^{2+}$-heme and kill rings and trophozoites via alkylation of proximal proteins, lipids, and heme[2–4]. *P. falciparum* resistance to ART, which first emerged in western Cambodia, is now present across the Greater Mekong Sub-region (GMS), compromising clinical efficacy[5–7]. Clinically, resistance is defined as a >5.5 h parasite clearance half-life (i.e. the time required to halve the parasite biomass)[8,9]. This reduced rate of clearance has increased the selective pressure on the partner drug. The recent appearance and spread across the GMS of resistance to the first-line partner drug piperaquine in ART-resistant parasites is causing widespread treatment failures, threatening local malaria control efforts[9–11]. A spread of ART resistance in Africa would be devastating[12].

Genetic and clinical data, supported by gene editing experiments, have identified point mutations in K13 (also known as Kelch13) as the dominant causal determinant of ART resistance. The degree of resistance is modulated by the specific mutation, which tends to cluster geographically[7,13], as well as by the parasite genetic background[14–19]. With cultured parasites, resistance (or tolerance) is typically defined as >1% survival in the Ring-stage Survival Assay (RSA), in which young rings (0–3 h post invasion (hpi)) are exposed to 700 nM DHA for 4–6 h and their survival measured 3 days later as a percentage of mock-treated parasites[20]. K13 R539T confers the highest level of in vitro resistance across strains (19–49% survival, with a geometric mean survival of 27%, as determined with four gene-edited strains), and is quite widespread at a relatively low prevalence across Asia[21]. By comparison K13 C580Y, which is the dominant isoform in the GMS, gives a more moderate in vitro resistance phenotype (4–24% survival, with a geometric mean survival of 9.2%, as determined with five gene-edited strains)[16,22]. K13-independent ART resistance has also been documented in five field isolates originating from Cambodia or Senegal and in two independent selection or transfection studies with the 3D7 parasite line[23–26].

The biological role of K13 and the precise mechanism(s) by which its mutations elicit an enhanced survival response to ART remain enigmatic. Studies have suggested roles for reduced endocytosis of hemoglobin leading to less $Fe^{2+}$-heme and reduced drug activation in rings, an endoplasmic reticulum (ER) stress response and an up-regulated unfolded protein response, translational arrest possibly involving enhanced eIF2-α phosphorylation, decreased levels of protein ubiquitination and altered rates of proteasome-mediated protein turnover, and PI3K-dependent intracellular signaling linked to amplified PI3P vesicles[27–35]. These processes could offset widespread ART-mediated damage to parasite proteins and other biomolecules[36,37]. One pressing question is whether these cellular effects are a direct result of mutations in K13, or whether they are epistatic phenomena that provide suitable cellular contexts on which mutant K13 can exert ART resistance, or a combination of both.

In this work, to explore the impact of K13 mutations on the parasite and gain further insight into the mechanism of ART resistance, we subjected a panel of *k13* gene-edited isogenic parasites on two different genetic backgrounds to untargeted multi-omics including transcriptomics, proteomics, and metabolomics (Fig. 1). Results provided herein reveal a broad array of intracellular processes affected by K13 mutations and converge

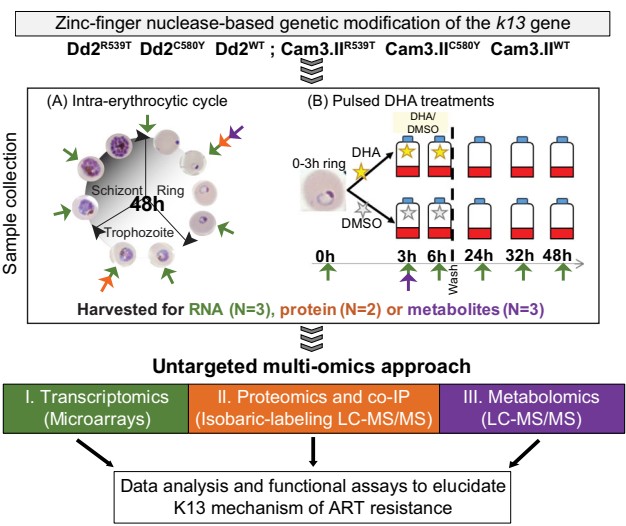

**Fig. 1 Outline of experimental workflow and multi-omics approaches applied to examine K13 function in this study.** The workflow illustrates the sample collection, processing, and data acquisition steps. The Dd2 laboratory-adapted parasite line and the Cam3.II clinical isolate were genetically modified at the *k13* locus via zinc-finger nuclease-based gene editing to create isogenic lines with either the wild-type (WT) or mutant *k13* alleles[15]. RNA, protein, or metabolite samples were collected for each parasite line at 6–7 time points throughout the 48 h asexual blood stage cycle in the absence of drug treatment. Synchronized K13 mutant and WT parasites were also subjected to a pulse of DHA, either at a concentration of 350 nM for 3 h (for metabolomics) or at 700 nM for 6 h (for transcriptomics), in parallel with 0.1% DMSO vehicle control. Samples were collected during DHA exposure and post drug removal. Colored arrows represent times and conditions of harvest for the different omic methods (green for transcriptomics, orange for proteomics and purple for metabolomics). Using microarray gene expression profiling and quantitative mass spectrometry-based proteomics and metabolomics, we generated global transcriptional, proteomic, and metabolomic profiles of gene-edited lines and compared the K13 mutants to their isogenic WT counterparts across the IDC.

on a previously unsuspected role for mitochondria. These data reveal vulnerabilities that could be leveraged to target ART-resistant *P. falciparum*.

## Results

**K13 mutations rewire the *P. falciparum* transcriptome.** To investigate the biological function of K13, we examined the transcriptional profiles of highly synchronized K13 mutant or isogenic wild-type (WT) parasite lines generated in the pre-ART era Dd2 and contemporary Cam3.II SE Asian strains[16] (Dd2R539T, Dd2C580Y, Dd2WT, Cam3.IIR539T, and Cam3.IIWT) across the 48 h IDC time course (Fig. 1, Table 1 and Supplementary Fig. 1a). Between *k13* genotypes, the Dd2 and Cam3.II sets of parasites mapped to highly similar IDC time points, with the isogenic sets of lines differing in developmental age by up to 1.5 and 4 h, respectively at the 0 h start of sampling. Principal component analyses of transcriptomic profiles showed distinct clustering by parasite stage (0–16 hpi rings vs. 24–40 hpi trophozoites and schizonts for PC1; trophozoites vs. schizonts for PC2), followed by genetic background (PC3) (Fig. 2a). Variation between strains outweighed the impact of K13 mutations on the transcriptomes, underscoring the need to compare isogenic edited parasites in order to identify changes specifically attributable to *k13* sequence variants.

**Table 1 Geographic origin, k13 isoform, and DHA response of *P. falciparum* lines.**

| Parasite | Edited? | K13 | Origin (year) | PCt$_{1/2}$ (h) | RSA (% survival) | Omics method |
|---|---|---|---|---|---|---|
| Cam3.II$^{R539T}$ | No | R539T | Pursat, W. Cambodia (2010) | 6.0 | 40.2 ± 3.0 | Transcriptomics, proteomics, co-IP |
| Cam3.II$^{C580Y}$ | Yes | C580Y | Pursat, W. Cambodia (2010) | N/A | 24.1 ± 2.9 | Proteomics, co-IP, metabolomics |
| Cam3.II$^{WT}$ | Yes | WT | Pursat, W. Cambodia (2010) | N/A | 0.3 ± 0.1 | Transcriptomics, proteomics, co-IP, metabolomics |
| Dd2$^{R539T}$ | Yes | R539T | Indochina (1980) | N/A | 19.4 ± 0.8 | Transcriptomics |
| Dd2$^{C580Y}$ | Yes | C580Y | Indochina (1980) | N/A | 4.1 ± 0.4 | Transcriptomics |
| Dd2$^{WT}$ | Yes (bsm) | WT | Indochina (1980) | N/A | 0.3 ± 0.1 | Transcriptomics |

*bsm* binding-site mutant control (no amino acid substitutions), *co-IP* co-immunoprecipitation, *N/A* not available, *PCt$_{1/2}$* parasite clearance half-life, *RSA* Ring-stage Survival Assay conducted with 0–3 h post invasion rings, *WT* wild type. Cam3.II and Dd2 parasites both harbor the PfCRT Dd2 haplotype: M74I/N75E/K76T/A220S/Q271E/N326S/I356T/R371I. Dd2 and Cam3.II harbor the PfMDR1 haplotypes N86Y and Y184F, respectively.

Because K13 mutations have been previously linked to prolonged duration of ring-stage development[29,38], we used Spearman rank correlations to map parasite ages to a high-resolution 96-time point reference transcriptome (Supplementary Fig. 1b–f). While the R539T mutants and WT parasites progressed similarly in both Dd2 and Cam3.II parasite backgrounds, the Dd2$^{C580Y}$ mutant developed ~4 h slower, most evident at the 24, 32, and 48 h time points (***$P < 0.001$, *t*-test, $N = 3$) (Fig. 2b and Supplementary Fig. 2a). This finding suggests that Dd2$^{C580Y}$ has a longer ring phase with a delayed ring to trophozoite transition.

At each sampling time point across the 48 h IDC, we observed ~80–200 genes differentially expressed (DE) between the Dd2$^{R539T}$ mutant line and its isogenic Dd2$^{WT}$ counterpart ($P < 0.05$, *t*-test, $n = 3$). Similar numbers of DE genes were observed in Dd2$^{C580Y}$ rings, with even higher numbers of DE genes observed at later time points (24, 32, and 48 h; Supplementary Fig. 2b and Supplementary Data 1), due to the decelerated development of Dd2$^{C580Y}$ parasites. A significant overlap of 394 genes were DE in the Dd2$^{R539T}$ and Dd2$^{C580Y}$ lines relative to Dd2$^{WT}$ parasites. Of these genes, 123 and 113 were up- and down-regulated, respectively, in both mutants within the same 4 h developmental window of the IDC (Fig. 2c). *K*-means clustering of the 394 DE genes showed a similar pattern of differential expression elicited by both mutants, indicating that the two different K13 mutations impacted expression of these genes in the same manner (Fig. 2c).

Most of these DE genes in Dd2 parasites were up-regulated during the transition from late schizonts to early rings and were enriched for pathways specialized in ubiquitin transfer (including ubiquitin-conjugating enzyme E2 and HECT-domain ubiquitin-transferase), protein phosphorylation, intracellular signaling (including adenylyl cyclase and cAMP protein kinase), sphingolipid and ceramide lipid metabolism, and mitochondrial electron transport chain (ETC) NADH dehydrogenase ubiquinone activity (hypergeometric test; $P < 0.05$) (Fig. 2d; Supplementary Fig. 2d and Data 1 and 2).

These same DE pathways were also observed in the Cam3.II$^{R539T}$ mutant. Interestingly, *k13* was among this set of genes with higher transcript abundance in the K13 mutant early rings. Functional enrichment analysis of DE genes at each sampling time point through the IDC for Dd2 and Cam3.II identified other strain-transcending changes associated with K13 mutations, ranging from elevated histone modifications in late rings, to higher levels of pyruvate metabolic enzyme, phosphoenolpyruvate (PEP) carboxykinase, and exported parasite proteins to the host red blood cell (RBC) cytosol in trophozoites, and increased *S*-nitrosylation and palmitoylation-modified proteins in schizonts (Fig. 2d and Supplementary Data 2). Several pathways were also exclusive to K13 mutants in Dd2 but not Cam3.II, including the DNA damage checkpoint (GO process) that contained genes that were up-regulated in ring stages in both Dd2 K13 mutant lines compared to Dd2$^{WT}$.

**K13 mutations alter the parasite proteome including protein turnover and glutamate and purine metabolism.** In parallel, we performed isobaric labeling-based quantitative proteomics with Cam3.II lines (Cam3.II$^{R539T}$, Cam3.II$^{C580Y}$, and Cam3.II$^{WT}$), and detected a total of 2780 and 2573 proteins (~50% of the *P. falciparum* proteome) in synchronized rings and trophozoites, respectively ($N = 2$; Fig. 1 and Supplementary Fig. 3a). Strikingly, fewer proteins were DE between K13 mutant and WT parasites in trophozoites, as compared to early rings that manifest ART resistance (Fig. 3a and Supplementary Fig. 3b, b). In rings, 87 proteins were more abundant in both K13 mutants relative to the isogenic WT line, in at least one experiment ($P < 0.05$; *t*-tests using normalized spectral intensity of peptides for each protein; Fig. 3a and Supplementary Data 3). Functional enrichment analyses converged on chaperone-mediated protein folding (cyclophilin 19B, ER calcium-binding protein, chaperonin 10 kDa) and protein targeting processes in the proteasome core complex (proteasomal subunits and ubiquitin ligase), redox (superoxide dismutase), and processes within intracellular vesicles and the digestive vacuole (Fig. 3a).

Of note, cyclophilin 19B was the top hit associated with in vivo ART resistance from an earlier transcriptomics study of *P. falciparum* clinical isolates[29]. Our observations here suggest that its increased protein abundance is regulated by K13 mutations. We also observed new functionalities not previously associated with K13, including carbon metabolism (illustrated by the increased abundance of glutamine synthetase and NADP-specific glutamate dehydrogenase that both help regulate cellular glutamine levels), as well as adenosine deaminase and hypoxanthine-guanine phosphoribosyltransferase (HGXPRT), which are critical for purine metabolism (Supplementary Data 3). While the changes associated with K13 mutations in rings were mostly instances of increased abundance, all 13 DE proteins in trophozoites showed reduced levels in the K13 mutants (Supplementary Fig. 3c). This included 1.8- to 2.0-fold less K13 protein in Cam3.II$^{C580Y}$ and Cam3.II$^{R539T}$ trophozoites (Fig. 3b), as previously reported[39]. However, unlike Cam3.II$^{R539T}$ rings that showed 1.5-fold reduced K13 protein, there was no detectable difference in Cam3.II$^{C580Y}$ rings (Fig. 3b). In these samples, K13 was identified with 11 unique peptides, covering 17% of the protein (Supplementary Fig. 4). The increased turnover of K13 protein in Cam3.II$^{R539T}$ rings may contribute to this parasite line's relatively high degree of ART resistance and suggests a potential loss-of-function role for the R539T mutation.

**Targeted metabolomics reveals altered levels of tricarboxylic acid (TCA) cycle and purine salvage metabolites in K13 mutants.** Metabolomic analyses of Cam3.II$^{C580Y}$ and Cam3.II$^{WT}$ ring ($N = 3$) and trophozoite ($N = 1$) extracts, using a previously published dataset[40], detected a total of 96 metabolites

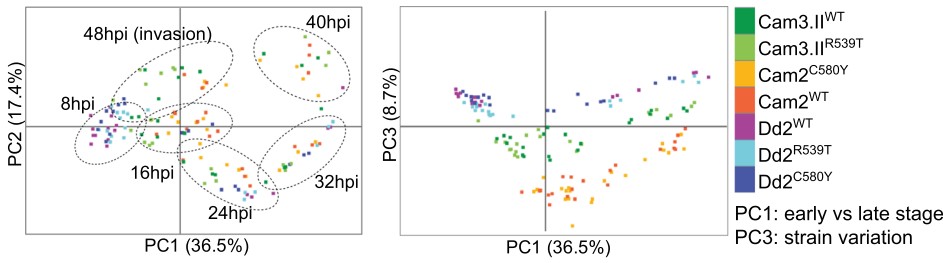

**a** Principal component analyses of K13 mutant and isogenic WT transcriptomes

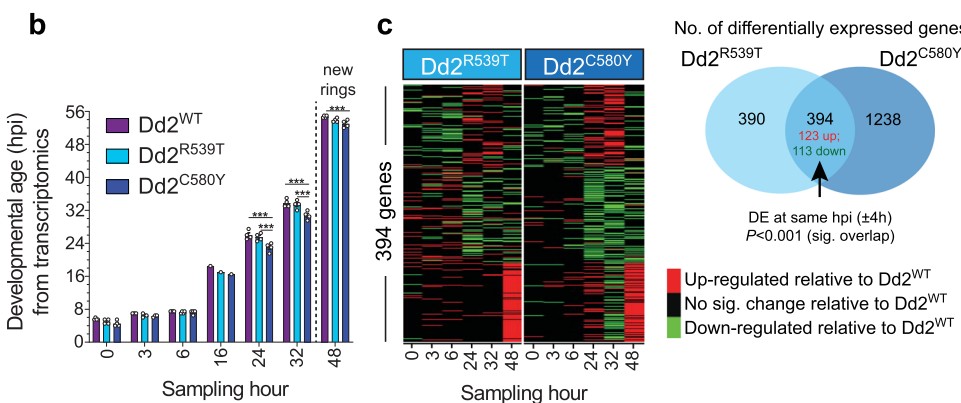

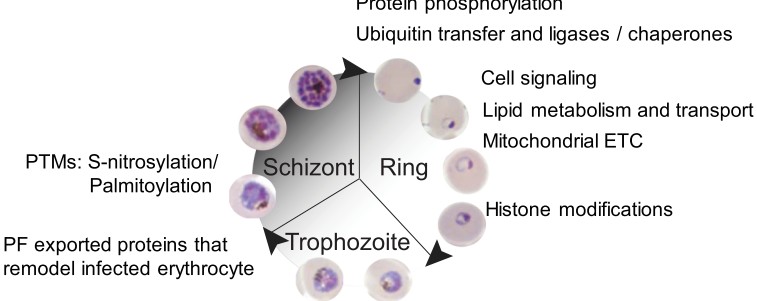

**d** Up-regulated gene sets in K13 mutant vs. isogenic WT parasites in Dd2 and Cam3.II lines

**Fig. 2 K13 mutations associate with altered temporal cell-cycle progression and broad transcriptional changes throughout the 48 h IDC. a** PCA plots for transcriptome samples labeled by parasite line, K13 status, sampling time, and parasite asexual stage. This analysis was conducted on Cam3.II and Dd2 K13 WT, C580Y and R539T samples, as well as the Cam2 isogenic K13 C580Y and WT pair[16]. **b** Transcriptomics-based age estimation for R539T and C580Y mutants and isogenic WT Dd2 parasites determined by comparison against a Dd2 reference 48 h IDC[68]. Dd2$^{C580Y}$ mutant parasites showed a significant albeit modest (up to 4 h) slowing of the developmental cycle during the 16–48 h developmental period, relative to Dd2$^{WT}$ and Dd2$^{R539T}$ parasites. Each time point was harvested on three independent occasions (data points are presented as means ± SEM), except for the 16 h time point that was collected only once. Differences between lines were examined using a two-sided t-test; ***$P < 0.001$. $P$ values for each comparison were 24 h time point–Dd2$^{C580Y}$ vs. Dd2$^{WT}$ $P = 1.7E−6$, Dd2$^{C580Y}$ vs. Dd2$^{R539T}$ $P = 5.1E−5$; 32 h time point–Dd2$^{C580Y}$ vs. Dd2$^{WT}$ $P = 1.7E−6$, Dd2$^{C580Y}$ vs. Dd2$^{R539T}$ $P = 2.6E−5$; 48 h time point–Dd2$^{C580Y}$ vs. Dd2$^{WT}$ $P = 0.002$. **c** Venn diagram (right) showing significant overlap of 394 DE genes in pair-wise comparisons between the gene-edited K13 mutant and WT lines at the same developmental stage (± 4 h). Heat map of k means-clustered overlapping DE genes in each pair-wise comparison, showing a similar profile of differential expression in both the R539T and C580Y mutants as compared to the WT line. Significant changes in expression levels were defined using a two-sided t-test ($P < 0.05$). Results also revealed consistent up-regulation of 80 genes in the 48 h late segmenter/early ring-stage samples of both K13 mutant parasite lines (see lower rows; detailed in Supplementary Data 1). The Venn diagram analysis used a binomial distribution test, with $P = 5.7E−32$. **d** Significantly up-regulated gene sets common to both K13 mutants relative to WT parasites (hypergeometric testing, $P < 0.05$) across the 48 h cycle in both the Cam3.II and Dd2 backgrounds (see Supplementary Data 2). ETC electron transport chain, hpi hours post invasion, PEP phosphoenolpyruvate, PTMs post-translational modifications.

(Supplementary Fig. 5a and Supplementary Data 4). Sample-to-sample correlation of the metabolite expression profiles showed that ring-stage samples clustered separately from trophozoites and uninfected RBC controls, indicating substantial changes in parasite metabolism throughout the IDC (Supplementary

Fig. 5b). Metabolomic set enrichment analyses of untreated ring-stage parasites revealed that the C580Y mutation affected the basal levels of metabolites involved in the TCA cycle, as well as purine, glutamate, and pyruvate metabolism (Fig. 3c). Partial least-squares discriminant analyses revealed several metabolites

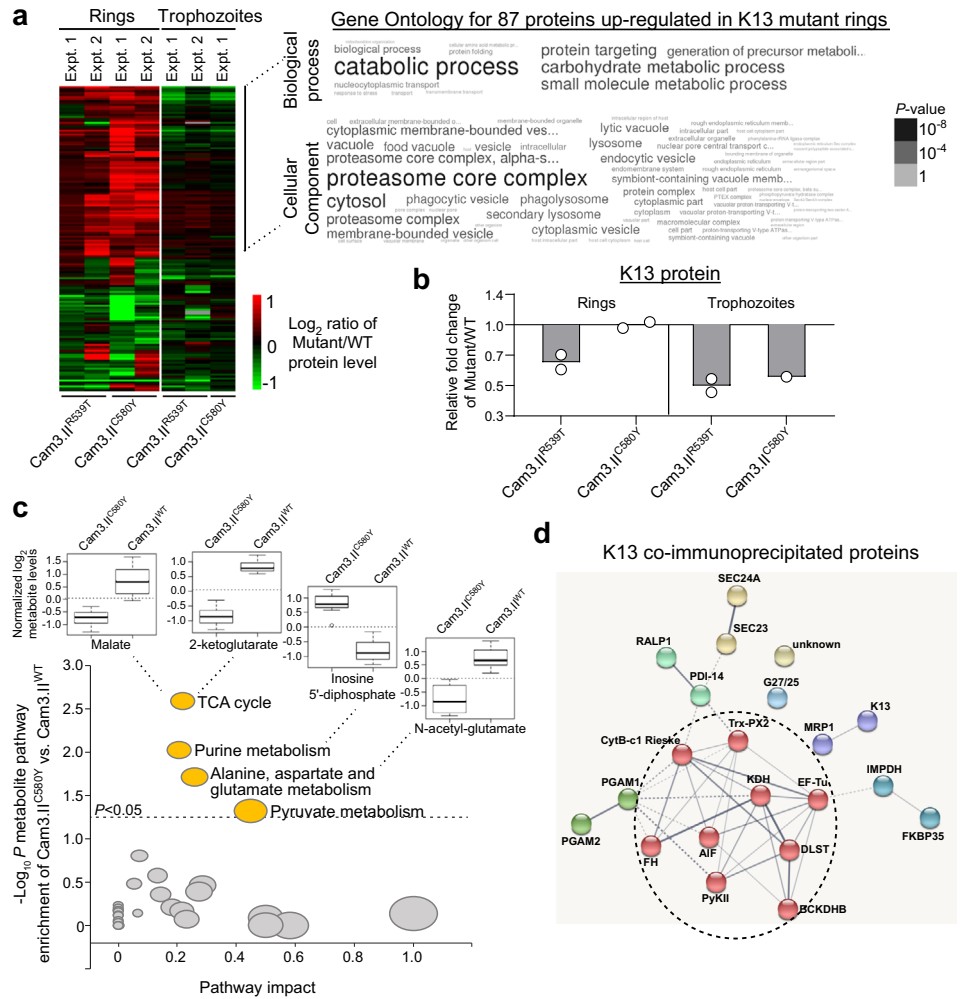

**Fig. 3 K13 mutations associate with differential expression of proteins involved in proteasome-mediated turnover and protein folding, and altered levels of TCA cycle, purine, glutamate, and pyruvate metabolites at the ring stage. a** Heat map of the $\log_2$ fold change of significantly different protein levels in Cam3.II$^{R539T}$ or Cam3.II$^{C580Y}$ mutants relative to the isogenic Cam3.II$^{WT}$ parasites for 155 DE proteins identified from isobaric labeling-based quantitative proteomics. Within each biological replicate, two-sided $t$-tests were performed for peptide levels belonging to each protein between K13 mutant and WT isogenic lines. Word cloud representation of functional analysis of significant biological processes and cellular components (Gene Ontology database) for the 87 up-regulated proteins in K13 mutant rings, along with their corresponding $P$ values. **b** Relative fold change of K13 protein levels in Cam3.II$^{R539T}$ and Cam3.II$^{C580Y}$ mutants vs. Cam3.II$^{WT}$, at the ring and trophozoite stages. K13 protein was ~2-fold lower in the Cam3.II$^{R539T}$ rings and trophozoites, as compared to the isogenic Cam3.II$^{WT}$ line. Data points are shown with means of the averaged normalized peptide abundance across two independent experiments for each strain and stage, except for C580Y trophozoites. **c** Scatterplot of the enrichment $-\log_{10} P$ score (using a two-sided $t$-test adjusted for multiple comparisons using the Holm–Bonferroni method) of the metabolite pathway and % pathway impact calculated from metabolite set enrichment analysis comparing Cam3.II$^{C580Y}$ and Cam3.II$^{WT}$ rings at basal level. Differential expression of four pathways ($P < 0.05$; gold color) is shown. Circle sizes reflect the pathway impact value, which represents the cumulative percentage of matched metabolite nodes (i.e. the importance measure) with respect to the total pathway based on pathway topological analyses. Box plots show reduced normalized $\log_2$ levels of the two TCA cycle metabolites, malate, and 2-ketoglutarate, and the glutamate metabolite, $N$-acetyl glutamate, but increased level of the purine metabolite, inosine 5′-diphosphate in Cam3.II$^{C580Y}$ as compared to Cam3.II$^{WT}$ rings. Data are represented as box and whisker plots with medians, interquartile ranges, and 95% confidence intervals ($N = 3$ independent experiments). **d** Interactome network of the 21 putative K13-interacting partners detected by co-immunoprecipitation (co-IP) with the K13-specific monoclonal antibodies E3 and D9 (ref. [41]) and mass spectrometry experiments using K13 WT, C580Y, or R539T samples from either the Cam3.II or CamWT strain background[16]. Proteins were clustered using the Markov Cluster Algorithm in STRING. Line thickness depicts the confidence/strength of the relationship between proteins (nodes). Solid lines represent intra-cluster interactions while dashed lines represent the interaction between clusters. Red nodes correspond to mitochondrial proteins, which are over-represented in this list of co-IP proteins. Gene abbreviations are listed in Supplementary Data 5.

with differential levels between the mutant and WT parasite (Supplementary Fig. 5c, d). The Cam3.II$^{C580Y}$ parasite had lower levels of malate and 2-ketoglutarate, both of which serve as TCA cycle intermediates, and $N$-acetyl glutamate, a glutamate metabolite, consistent with higher NADP-specific glutamate dehydrogenase and glutamine synthetase RNA transcripts in the

mutant lines (Fig. 3c). In contrast, we observed higher levels of the nucleoside precursor inosine 5′-diphosphate and pyruvate metabolite phosphoenolpyruvate (PEP). These findings were consistent with higher HGXPRT protein abundance and elevated PEP carboxykinase RNA transcript levels (Fig. 2d and Supplementary Figs. 2d and 5d, e). Collectively, these data suggest a

higher reliance on reverse glutaminolysis, with alterations in glycolysis/gluconeogenesis, the mitochondrial TCA cycle, and purine metabolism in the K13 mutants.

**Co-immunoprecipitation assays identify mitochondrial factors as major putative K13-interacting partners.** To identify putative interactors of K13 that could drive these RNA, protein, or metabolomic changes described above, we earlier performed co-immunoprecipitation (co-IP) and mass spectrometry experiments using custom-raised K13-specific monoclonal antibodies on *k13*-edited and WT Cambodian isolates. The specificity of these antibodies was validated using western blot and IFA data comparing our K13 signals with results from parasite lines expressing GFP-K13 or 3×HA tagged K13 (ref. [41]). These experiments yielded 21 proteins that were present in at least half of the six co-IP experiments, and absent in four negative resin controls. Interactome analyses using the STRING database, which uses published biological data to search for observed and predicted protein–protein interactions[42], predicted a statistically significant enrichment for interactions among these proteins (protein–protein interaction *P* value of 3.1E−7) (Fig. 3d and Supplementary Data 5). Besides identifying physical associations of K13 with chaperones involved in protein folding and vesicle trafficking, we also observed an over-representation of mitochondrial-localized proteins (in 9 of 21 proteins; Fig. 3d, red nodes). These proteins are important for a range of mitochondrial functions including the TCA cycle (fumarate hydratase and multiple 2-oxoglutarate dehydrogenase components), the ETC (cytochrome $bc_1$ complex), translation (elongation factor Tu), Fe–S protein synthesis (ferrodoxin reductase-like protein), and antioxidant processes (thioredoxin peroxidase 2). This evokes a potential functional interplay between mitochondrial metabolism, signaling, and the unfolded protein response in *P. falciparum*.

**Metabolomic and transcriptional response of K13 parasites to DHA.** We next applied metabolomics and transcriptomics approaches to explore the response of K13 lines to DHA exposure. Metabolomic profiling of parasites after a 3 h treatment with 70 or 350 nM DHA revealed a substantial reduction in the levels of several hemoglobin-derived peptides (including LD, SD, PE, PD, PEE, DLS, DLH, and PVNF) in Cam3.II$^{C580Y}$ and Cam3.II$^{WT}$ (Fig. 4a). This reduction was of a similar magnitude in both Cam3.II K13 mutant and WT lines at ring and trophozoite stages. In the absence of DHA, there was no difference in hemoglobin-derived peptides levels between Cam3.II$^{C580Y}$ and Cam3.II$^{WT}$ rings (Supplementary Fig. 5f). This finding suggests that the reduction in hemoglobin-derived peptides upon treatment with DHA occurs via a K13-independent mechanism, involving either the inhibition of hemoglobin uptake or its digestion by the parasite.

**Mutant K13 promotes recovery after initial cell-cycle arrest in DHA-treated parasites.** To further characterize the effects of DHA on cell-cycle progression and the protection afforded by K13 mutations, early ring-stage parasites were subjected to a 6 h pulse of 700 nM DHA or 0.1% DMSO vehicle control and samples were collected during and after DHA exposure (Fig. 1). Transcriptional profiling and age mapping of WT and mutant lines ± DHA using Spearman rank correlations revealed that DHA-sensitive K13 WT parasites were completely halted in their developmental progression over the period of 6–48 h post DHA treatment (Fig. 4b, c, see solid purple and green lines, and Supplementary Fig. 1b–f). In contrast, DHA-resistant K13 mutants stalled for up to 16 h from the start of treatment and then resumed normal cell-cycle development asynchronously (Fig. 4b,

c—see solid aqua, blue, and orange lines). The speed of re-initiation of transcriptional activity correlated with the degree of parasite's resistance to ART afforded by the K13 mutation in the Dd2 and Cam3.II backgrounds (Fig. 4b, c). Developmental differences were independently verified by Giemsa smears. Morphologically, we observed that DHA-treated K13 WT parasites failed to develop beyond rings, with pyknotic forms seen 24 h post DHA treatment. In contrast, a subset of DHA-treated K13 mutant parasites recovered after the initial drug-induced developmental arrest, progressed to schizonts, and reinvaded RBCs to form new rings (Fig. 4d and Supplementary Fig. 6).

**Differential transcriptional response of K13 mutants and WT lines to DHA treatments.** Compared to the transcriptional impact of K13 mutations, DHA treatment affected a much larger number of genes, up to 2530 and 2517 in K13 WT Dd2 and Cam3.II parasites respectively (at *P* < 0.05; with cutoffs of >1.4-fold for either Dd2 or Cam3.II). These transcriptional changes occurred mostly at 32–40 h post treatment (Supplementary Fig. 7). Gene set enrichment analyses with Dd2$^{WT}$ rings revealed that pathways relating to protein metabolism such as protein ubiquitination, degradation, folding, and translation, as well as redox and glycolysis, were significantly down-regulated within 6 h of initiating DHA treatment (*P* < 0.05 and false discovery rate <0.3; Supplementary Data 6). Our observations agree with a prior report that DHA treatment of K13 WT drug-sensitive parasites caused a rapid drug-triggered shutdown in critical cellular processes[43]. We note that fewer genes were DE in K13 mutants vs. WT parasites after DHA exposure and the transcriptional changes in these genes were attenuated (Supplementary Fig. 7).

To identify genes that associate with the survival of post DHA-pulsed K13 mutant parasites, we applied two-factor ANOVA on transcriptional responses in the Dd2 and Cam3.II lines, stratified by K13 status (mutant vs. WT) and treatment condition (DHA vs. DMSO control). In total, 667 and 192 genes were DE from 3 to 48 h post initiation of DHA treatment in Dd2 or Cam3.II mutants vs. their WT counterparts respectively (Fig. 4e and Supplementary Data 7). We observed a larger divergence in transcriptional response to DHA between Dd2$^{R539T}$ and Dd2$^{WT}$ than between Dd2$^{C580Y}$ and Dd2$^{WT}$ (Fig. 4e), consistent with the higher survival rate in the R539T mutant compared with C580Y. Further analyses of the initial 3 and 6 h post DHA/DMSO transcriptional responses in Dd2$^{R539T}$ parasites that associated with elevated survival in the early ring stage identified 348 and 234 genes with higher and lower RNA expression, respectively, in the DHA-treated Dd2$^{R539T}$ mutant relative to the Dd2$^{WT}$ line (*t*-test; *P* < 0.05) (Fig. 4f). Functional enrichment analyses revealed highly similar up-regulated gene sets at the initial 6 h time point and throughout the 48 h sampling, including Rab-mediated vesicular trafficking (Rab 1A, 5B, and 11A), secretory and endocytic complexes, organellar transcription, and mitochondrial ATP production (Fig. 4f and Supplementary Fig. 8a). These pathways were also observed in analyses of DHA-treated Cam3.II$^{R539T}$ vs. WT lines (Supplementary Fig. 8a, indicated with #, and Supplementary Data 7). Gene sets specific to 6 h response comprise tRNA modifications and lipid transport and those at 48 h in both backgrounds were limited to redox and DNA excision repair. In contrast, fewer commonalities were seen between down-regulated genes in DHA-treated K13 mutants as compared to WT parasites, in both parasite backgrounds (Supplementary Fig. 8b and Supplementary Data 7). Genes down-regulated in DHA-treated Dd2$^{R539T}$ vs. Dd2$^{WT}$ from 6 to 48 h were enriched for protein folding and ER heat-shock chaperones, autophagy markers (ATG7 and ATG18), and phosphorylated proteins (Fig. 4f). At the 6 h time point, we observed down-regulation of

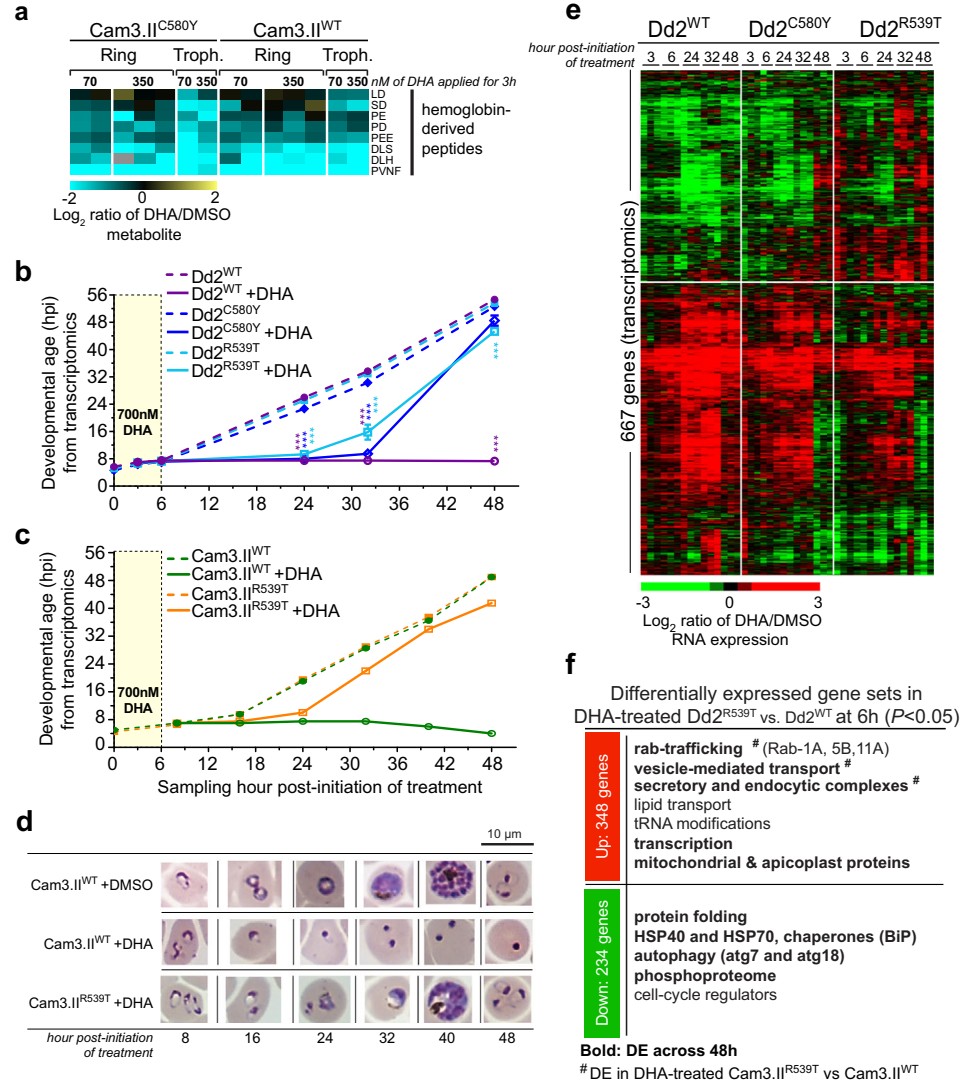

**Fig. 4 Short-term DHA exposure results in a decrease of hemoglobin-derived peptides, and a temporary, reversible cell-cycle arrest in K13 mutant parasites, concomitant with distinct transcriptional responses between K13 mutant and WT parasites. a** Heat map of $\log_2$ fold changes in DHA-treated Cam3.II$^{C580Y}$ or Cam3.II$^{WT}$ parasites vs. DMSO controls for the eight metabolites that showed a significant change in level after parasites were pulsed with either 70 or 350 nM DHA for 3 h. DHA treatment at rings and trophozoites resulted in a consistent down-regulation of host hemoglobin-derived peptides in both lines. For levels in untreated rings, see Supplementary Fig. 4f. **b**, **c** Transcriptome-based age estimation of the *k13*-edited R539T, C580Y, and WT in Dd2 or Cam3.II parasites after 6 h pulsed exposure to either 700 nM DHA (solid lines) or DMSO vehicle control (dashed lines). Parasites were sampled up to 48 h post treatment and developmental ages were determined using Spearman rank correlations. Results shown as means ± SEM ($N = 3$ for each Dd2 line and $N = 1$ for Cam3.II lines). ***$P < 0.001$, unpaired two-sided *t*-tests. *P* values in **b** are: 24 h time point—Dd2$^{WT}$ DHA vs. DMSO $P = 1.7\text{E}{-}5$, Dd2$^{C580Y}$ DHA vs. DMSO $P = 1.7\text{E}{-}5$, Dd2$^{R539T}$ DHA vs. DMSO $P = 4.7\text{E}{-}6$; 32 h time point—Dd2$^{WT}$ DHA vs. DMSO $P = 2.5\text{E}{-}6$, Dd2$^{C580Y}$ DHA vs. DMSO $P = 2.4\text{E}{-}6$, Dd2$^{R539T}$ DHA vs. DMSO $P = 0.002$; 48 h time point—Dd2$^{WT}$ DHA vs. DMSO $P = 3.7\text{E}{-}9$, Dd2$^{R539T}$ DHA vs. DMSO $P = 3.8\text{E}{-}6$. **d** Giemsa-stained parasite images showed that DHA-treated Cam3.II$^{WT}$ parasites became pyknotic after 24 h, whereas Cam3.II$^{R539T}$ parasites underwent a lag in progression through the IDC following DHA treatment, compared to DMSO mock-treated controls. These images are representative of the majority (>50%) of parasites observed at each time and condition per parasite line, with additional examples provided in Supplementary Fig. 6. Microscopy analyses were performed for at least 5000 cells counted across three independent experiments, which yielded similar results. **e** Heat map of $\log_2$ ratios of DHA-treated/DMSO-treated mRNA expression levels for the 667 hierarchically clustered genes whose expression differed significantly in the Dd2$^{C580Y}$ and/or Dd2$^{R539T}$ mutants as compared with Dd2$^{WT}$ parasites across 3–48 h post initiation of treatment ($N = 3$ independent experiments; two-factor ANOVA with adjusted $P < 0.05$; pathways listed in Supplementary Fig. 6 and Supplementary Data 7). A more divergent transcriptional response was observed with Dd2$^{R539T}$ parasites that display high levels of ART resistance in vitro compared with Dd2$^{C580Y}$ parasites[16]. **f** Functional enrichment analyses identified biological pathways significantly enriched ($P < 0.05$, hypergeometric tests) among the genes that were up- or down-regulated in the DHA-treated Dd2$^{R539T}$ as compared with Dd2$^{WT}$ parasites relative to DMSO controls in early rings after 3 and 6 h post DHA exposure. Up-regulated genes were relatively more abundant, reflecting increased transcript levels induced by DHA-treated Dd2$^{R539T}$ parasites vs. Dd2$^{WT}$ parasites as compared to their respective DMSO controls. Down-regulated genes reflected a reduction in transcription or faster turnover of these transcripts in DHA-treated Dd2$^{R539T}$ parasites vs. Dd2$^{WT}$ parasites, compared to their DMSO controls. Gene sets in bold or labeled as # were differentially expressed over the 48 h period of sampling or between DHA-treated Cam3.II$^{R539T}$ parasites vs. isogenic Cam3.II$^{WT}$ parasites respectively (see Supplementary Data 7 for pathways and genes).

cell-cycle regulators, which may be important for the parasite's exit from a temporary state of DHA-induced quiescence. Our data suggest that in WT parasites DHA causes a shutdown of energy-related processes. In contrast, processes including synthesis of ATP and vesicular transport across the ER, Golgi, and plasma membranes can remain transcriptionally active in resistant K13 mutant parasites even as early as 6 h post DHA exposure and can be sustained up to 48 h post DHA removal. Higher expression of membrane-bound transporters and the ATP synthase complex subunits might reflect the enhanced ability of K13 mutants to maintain their intracellular pH gradients and ATP levels following DHA-induced oxidative stress.

**Multi-omics results associate K13 with mitochondria-related energetic processes.** Given our multi-omics results that linked K13 to mitochondrial and purine metabolism (Fig. 2d and Fig. 3a, c, d), we assessed the susceptibility of K13 mutant and WT parasites to a panel of mitochondrial and purine metabolism inhibitors (Fig. 5a). Dose–response assays showed that Cam3.II$^{R539T}$ and Cam3.II$^{WT}$ parasites had equivalent sensitivities to inhibitors of mitochondrial protein synthesis (fusidic acid), purine metabolism (immucilin G and H and ribavirin), or pyrimidine metabolism (the DHODH inhibitor DSM265). These results were

obtained in both 4 h assays that measure activity against early rings and 72 h standard assays (Fig. 5b and Supplementary Fig. 9a). In contrast, with the slower-acting mitochondrial ETC inhibitor atovaquone[44,45] (ATQ), we noted a subtle yet statistically significant 1.6-fold increased sensitivity in Cam3.II$^{R539T}$ rings as compared with isogenic Cam3.II$^{WT}$ rings (shown as lower IC$_{50}$ and IC$_{90}$ values in Fig. 5b). Individual IC$_{50}$ and IC$_{90}$ data are provided in Supplementary Data 8.

**K13 mutant parasites exhibit differential responses to atovaquone plus DHA.** We next performed drug combination assays by exposing tightly synchronized early ring-stage parasites to varying concentrations of both DHA and ATQ, as determined by molar ratios of their individual IC$_{50}$ values. Dose–response curves with early rings exposed for 4 h to increasing ratios of ATQ:DHA showed evidence of increased ATQ sensitization of Cam3.II$^{R539T}$ but not Cam3.II$^{WT}$ parasites to DHA (Fig. 5c, d). At a 4:1 ATQ: DHA ratio, the Cam3.II$^{R539T}$ parasites became fully sensitive to DHA. Isobologram analyses of the 4 h pulsed ATQ and DHA drug-pair combination assays revealed moderate synergy in Cam3.II$^{R539T}$, whereas an additive interaction was seen in the DHA-sensitive Cam3.II$^{WT}$ line as well as the Cam3.II$^{C580Y}$ line that was less resistant than the R539T mutant (Supplementary

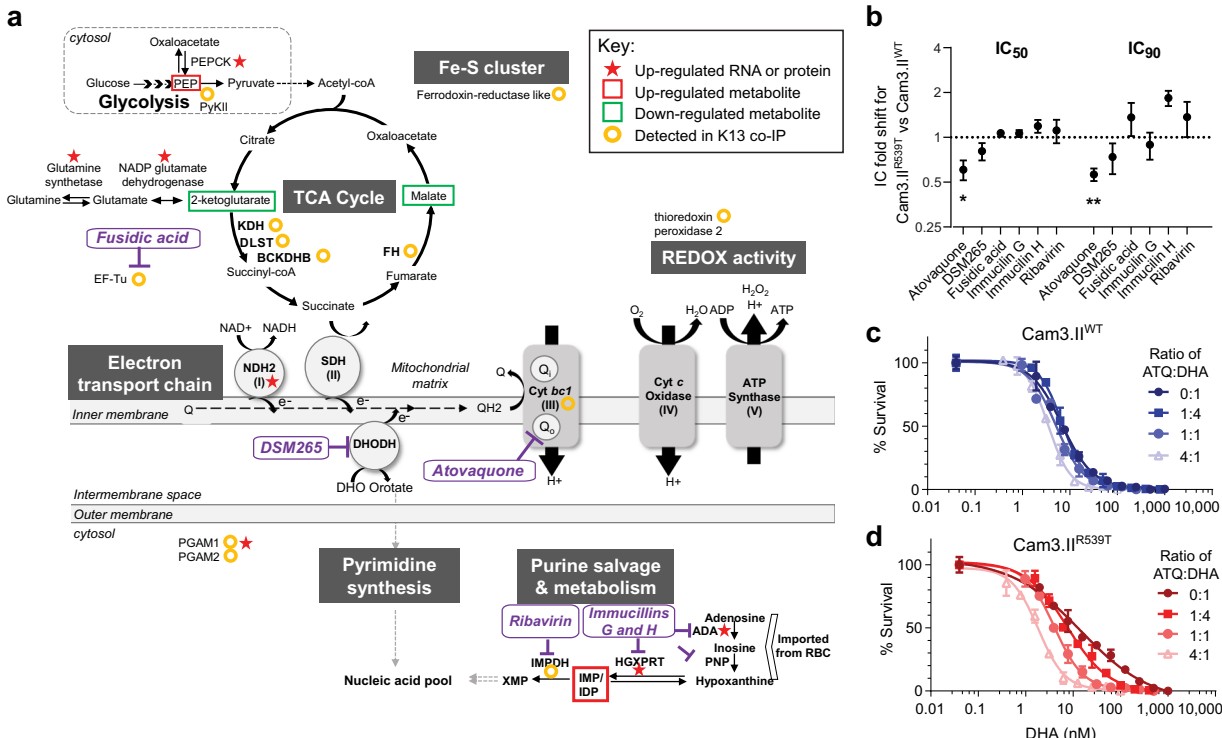

**Fig. 5 Multi-omics data and dose–response assays reveals K13's role in maintaining mitochondrial ETC functions. a** Schematic of the energy and purine metabolism pathways in the parasite mitochondrion. Genes, proteins, metabolites, and co-IP proteins that were detected as differentially regulated between isogenic K13 mutant and WT parasites in multi-omics experiments are highlighted. ADA adenosine deaminase, DHODH dihydroorotate dehydrogenase, DLST dihydrolipoyllysine-residue succinyltransferase, HGXPRT hypoxanthine-guanine-xanthine phosphoribosyltransferase, IMPDH inosine 5'-monophosphate dehydrogenase, PNP purine nucleoside phosphorylase. For full list of protein names, refer to Supplementary Data 5. **b** IC$_{50}$ and IC$_{90}$ fold shifts for Cam3.II$^{R539T}$ vs. Cam3.II$^{WT}$ 0–3 hpi early rings exposed for 4 h to a panel of inhibitors targeting mitochondrial processes (atovaquone, DSM265, fusidic acid) or purine metabolism (immucilin G and H, ribavirin). Shown are means ± SEM values from two to five independent experiments with technical duplicates. **P = 0.008, *P = 0.024, two-tailed one-sample t-tests of the IC fold shift. Individual IC$_{50}$ and IC$_{90}$ data are provided in Supplementary Data 8. **c** Four hour dose–response assays conducted on Cam3.II$^{WT}$ 0–3 hpi early rings, depicting no change in the DHA dose–response curves in the presence of increased ATQ to DHA ratios. **d** Four hour dose–response curves show increased sensitization of Cam3.II$^{R539T}$ 0–3 hpi early rings to DHA with higher ratios of ATQ to DHA. The mitochondrial ETC inhibitor ATQ displays increased potency when combined with DHA against Cam3.II$^{R539T}$ parasites in 4 h drug treatments. Dose–response assays were performed in three to five independent experiments with technical duplicates for each parasite line. The data presented in **c** and **d** are exemplars from one independent experiment with error bars representing the SD between technical duplicates. Also see Supplementary Fig. 9b–e for FIC$_{50}$ curves and isobologram analyses.

Fig. 9b). This synergy was reflected in a significantly lower Combination Index (the averaged sum of fractional $IC_{50}$ ($FIC_{50}$) values), in Cam3.II[R539T] compared to Cam3.II[WT] parasites (Supplementary Fig. 9b, inset). In 72 h assays, an antagonistic relationship between ATQ and DHA was observed, with its effect being most pronounced in WT parasites and less so in C580Y and especially R539T parasites (Supplementary Fig. 9c). These results were reproduced, albeit to a lesser degree, in Dd2 parasites that carry the same K13 genotypes and that are comparatively less DHA-resistant than the Cam3.II lines (Supplementary Fig. 9d, e). These results suggest that ATQ-mediated inhibition of the mitochondrial cytochrome $bc_1$ $Q_o$ site can reduce the recovery of K13 mutant ring-stage parasites following treatment with pro-oxidant ART drugs.

## Discussion

Our multi-omics analysis of isogenic *P. falciparum* lines expressing mutant or WT isoforms of the ART resistance determinant K13 reveal a striking array of physiological processes in asexual blood stage parasites that are uniquely altered by K13 mutations (Fig. 6). First, cell-cycle periodicity was differentially impacted. The parasite's developmental progression decelerated during the ring stage in the C580Y mutant but not in the R539T mutant. This contrasted with K13 protein levels that were not significantly altered in the C580Y mutant yet were lower in the R539T mutant. Recent reports highlight an association between K13 inactivation and lowered endocytosis, where less host hemoglobin is taken in and digested, resulting in less liberation of $Fe^{2+}$-heme and reduced ART activation[27,28,46]. Based on our findings, reduced endocytosis could stem from either lowered K13 protein levels and activity in the R539T mutant or prolonged ring development caused by the C580Y mutation. Upon DHA exposure, K13 mutant trophozoites were earlier reported to show net reductions in heme-DHA adducts, relative to WT counterparts[39,47]. Our

observation that fewer genes were DE in DHA-treated K13 mutant parasites, as compared to DHA-treated isogenic WT lines, supports the hypothesis that a lower level of activated ART accumulates and exerts its effect in K13 mutant parasites. However, this did not account for the increased rates of survival of DHA-treated rings (0–6 hpi) elicited by K13 mutations and the re-initiation of growth after a temporary arrest, implicating a role for additional survival mechanisms. Furthermore, in our study the reduction in hemoglobin-derived peptides upon treatment with DHA, observed within 3 h of drug exposure, occurred to a similar degree in both K13 mutant and WT parasites. These results suggest that the mechanism of ART resistance afforded by mutant K13 extends beyond a role for reduced hemoglobin endocytosis in rings.

Our transcriptomic analysis revealed that in the absence of DHA, nearly 400 genes were DE in Dd2[R539T] and Dd2[C580Y] mutant parasites as compared to isogenic Dd2[WT] parasites, across the 48 h IDC. The largest number of DE genes was observed in newly invaded rings, when K13 mutations impart ART resistance. Among these genes, most were up-regulated in the mutants. Gene set enrichment analysis revealed up-regulation of pathways involved in post-translational modifications (including phosphorylation and palmitoylation), protein export or turnover, lipid metabolism and transport, and the mitochondrial ETC. Many of these pathways were conserved between Dd2 and Cam3.II parasites, although individual genes often differed, suggesting that the genetic background influences the impact of K13 mutations on the parasite transcriptome. These data provide evidence for pleiotropic effects of mutant K13 that are common across parasite strains. Other loci must nonetheless be important in determining the extent to which mutant K13 mediates ART resistance. This is shown by the greater levels of RSA survival in Cam3.II parasites compared to Dd2 (40.2% vs. 19.4% for the two respective R539T mutants[16]), suggesting that K13 also has epistatic interactions with other modifier loci. Intriguingly, K13-mutant Asian parasites

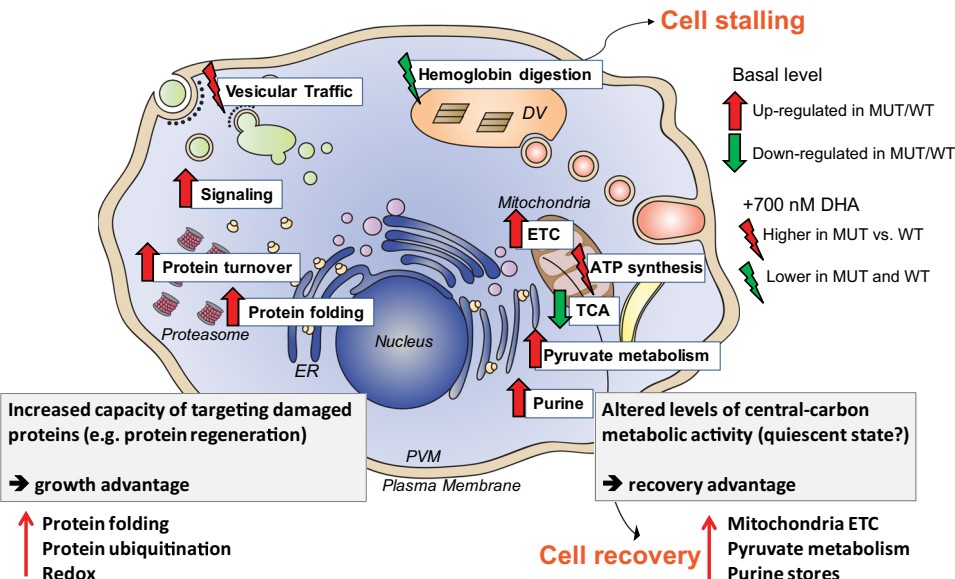

**Fig. 6 Proposed model for the role of K13 in ART resistance.** K13 mutations are associated with an up-regulation of protein folding as well as proteasome-linked protein turnover that may provide a growth advantage via a greater capacity of the mutant parasite to target and eliminate damaged proteins caused by the proteotoxic ART drug. K13 also appears to be involved in other novel pathways including intracellular signaling, pyruvate metabolism, purine salvage, and the mitochondrial electron transport chain (ETC). These functions suggest an altered central carbon metabolism and represent a quiescent state maintained by the parasite that allows for a recovery advantage following DHA exposure. DHA inhibits hemoglobin uptake or degradation and induces stalling in both K13 mutant (MUT) and wild-type (WT) rings. Recovery in K13 mutants may occur through enhanced mitochondrial ATP metabolic function and remodeling of the parasite's secretory and vesicular transport processes. Inhibition of mitochondrial ETC activity lowers the survival of K13 mutant parasites to DHA treatment.

were recently found to carry mutant alleles of DNA repair genes that provided enhanced protection against artesunate-mediated DNA damage; several of these genes are also present in Dd2 (ref. [48]). Other studies have also reported founder genetic backgrounds on which mutant K13 evolved in Southeast Asia, supporting a role for epistatic interactions that contribute to the resistance phenotype and that may help with a process referred to as cellular healing in ART-treated K13-mutant parasites[17,49,50].

K13 C580Y and R539T mutations showed very similar impacts on both parasite transcription and translation in untreated parasites. Transcriptomics and proteomics results overlapped in the up-regulation of chaperone-mediated protein folding and the ubiquitin–proteasome system, as observed in rings of ART-resistant K13-mutant clinical isolates sampled ex vivo[29]. The higher protein abundance of the protein folding factor cyclophilin 19B in K13 mutants reaffirms its earlier report as the top-ranked transcriptional marker in K13 ART-resistant isolates[29] and suggest its involvement in enhancing the parasite's capacity to survive the general proteotoxic effects of ART action. In considering these data, we note that experiments had a limited number of independent repeats (generally two) and did not include reference pools or labeling efficiency tests. High reproducibility was nonetheless observed between repeats (Fig. 3a and Supplementary Fig. 3b).

Our combined transcriptomics, proteomics and metabolomics data uncovered new functional modules associated with K13 mutants, including pyruvate and glutamate-linked carbon metabolism, purine metabolism, the mitochondrial ETC, and the TCA cycle (Figs. 5a and 6). Pyruvate and glutamate metabolism are highly connected with mitochondrial functions. Pyruvate metabolism is a core energetics pathway that provides acetyl-coA for the TCA cycle, histone acetylation, and lipid metabolism[51,52]. Pyruvate and aspartate synthesis can compensate for deficiencies in ETC and membrane potential generation in mammalian cells[53]. Glutamate is a branch point for carbon and nitrogen metabolism, feeding into the mitochondrial TCA cycle via its conversion into 2-ketoglutarate or into glutamine for nitrogen storage, in preparation for starvation conditions. Our findings of reduced levels of the mitochondrial TCA cycle intermediates malate and 2-ketoglutarate in K13 mutants are consistent with recent modeling predictions based on transcriptional analysis of ART-resistant versus -sensitive clinical isolates[54]. In rodent *Plasmodium* parasites, the addition of malate or succinate promotes oxygen consumption, which contributes to the mitochondrial membrane potential ($\Delta\Psi$) and stimulates complex III activity[55,56]. The lower levels of malate observed in our metabolomics data suggest that K13-mutant rings might mimic a state of anoxia and lower the ETC complex III activity. Collectively, these findings point towards a rewiring of the energy requirements and metabolic state in K13 mutant parasites, and suggest that a higher reliance on reverse glutaminolysis with altered glycolysis/gluconeogenesis, and lowered mitochondrial TCA cycle and ETC functions, might facilitate the recovery of DHA-treated K13-mutant rings. One question that merits further exploration is whether the reported impact of K13 mutations on reducing hemoglobin endocytosis[27,28] might trigger a semi-starvation state that accounts for some of our observed metabolic changes. A recent study showed that the Cam3.I^R539T Cambodian parasite line[16] showed a threefold better rate of recovery following 72 h of isoleucine starvation as compared with the isogenic Cam3.I^WT Cambodian parasite line, suggesting that K13-mutant parasites might be better able to tolerate semi-starvation[57]. Nonetheless, transcriptomic profiling of amino acid-starved 3D7 WT K13 parasites showed a general slowing of transcription across the IDC[58], without evidence of the specific pathways observed in our study. Metabolomics data in that study also showed minimal

overlap with our results, suggesting that mutant K13-mediated reduction in hemoglobin endocytosis does not readily mimic starvation responses resulting from amino acid deprivation.

Further evidence of mitochondrial-linked function in K13 mutants was seen in K13's physical interaction with mitochondrial factors, which may enable K13 parasites to survive the DHA pulse by reinitiating growth after a temporary 16 h arrest[41]. Co-IP experiments identified PGAM2, the parasite ortholog of PGAM5 that is a substrate of the mammalian ubiquitin ligase complex involving the K13 ortholog KEAP1 (refs. [59,60]). In mammalian systems, KEAP1 associates physically with NRF2 and PGAM5 on the outer mitochondrial membrane. Upon stress, the mitochondrial unfolded protein response occurs via dissociation of KEAP1 from the NRF2–PGAM5 protein complex, leading to retrograde signaling from the mitochondria to the nucleus and expression of antioxidant and stress response genes[60]. We also observed an increased expression of redox enzymes 48 h after DHA treatment in K13 mutant parasites. These findings evoke a potential functional interplay between mitochondrial metabolism, signaling and the unfolded protein response in *P. falciparum*. We note that our co-IP results differ from the set of K13-interacting proteins identified using a quantitative dimerization-induced bio-ID approach with GFP-tagged K13 (ref. [27]). Partial overlap was nonetheless observed between our study and a separate report that used GFP-Trap beads to affinity purify GFP-K13 and then identified the immunoprecipitated proteins[18]. These results merit further investigation using independent approaches such as co-IP data with antibodies specific to mitochondrial proteins, which we are to date unable to obtain.

Upon exposure to DHA, survival of K13 mutants was associated with a shutdown of core metabolic functions including protein synthesis, transcription, and glycolysis, accompanied by sustained elevated RNA expression of specific pathways including vesicle-mediated intracellular transport and ATP production. The association with vesicular transport recalls earlier associations between mutant K13 and amplification of PI3P-positive vesicles that were postulated to neutralize ART proteotoxicity[34]. The speed of re-initiation of transcriptional activity correlated with the degree of ART resistance afforded by K13 mutations in the Dd2 and Cam3.II backgrounds. We hypothesize that this might occur via the regulation of mitochondria-linked quiescence. This possibility is supported by the greater extent of sensitization to DHA by ATQ (an ETC complex III $Q_o$ ubiquinol oxidation inhibitor) in the Cam3.II mutants compared to the Dd2 K13 mutants (that impart lower resistance) or sensitive WT lines.

In the mitochondria of mammalian cells, redox equilibrium is regulated through functions including oxygen uptake, ATP production, maintenance of membrane potential, and generation of reactive oxygen species. Dysregulation of these processes can drive cells towards senescence or proliferation[52,61]. In certain cancerous states, cells can switch between senescence and proliferation by modulating their reliance on glycolysis vs. oxidative phosphorylation (involving the TCA cycle and the ETC)[52,62,63]. Recent research has described drug-induced "low glycolysis-low oxidative phosphorylation" phenotypes that enable mammalian cells to survive drug challenge[62]. A potential role for mitochondria in sensing ART action and regulating cellular quiescence was recently evoked in a study with *Mycobacterium tuberculosis*, which found that ART can inhibit intracellular signaling and persistence via adduct formation with heme-carrying kinases that act as redox or hypoxia sensors[64]. In *P. falciparum*, we also observed substantial mitochondrial oxidation within 4 h of DHA treatment, at levels considerably greater than those observed with either chloroquine or ATQ[41]. Our data suggest a metabolic rewiring in *P. falciparum* parasites, with a dysfunctional

mitochondrion that can be perturbed by combining ATQ with DHA. Interestingly, a genome-wide CRISPR/Cas9 screen in the Apicomplexan parasite *Toxoplasma gondii* recently identified components of the TCA cycle and heme biosynthesis as mitochondrial determinants of DHA susceptibility[65]. One important distinction, however, is that unlike *Plasmodium*, *Toxoplasma* parasites do not endocytose or metabolize hemoglobin, thereby removing this pathway for ART activation. Further studies are merited to assess whether altered mitochondrial metabolism might constitute a vulnerability that can be leveraged to overcome ART resistance.

In summary, our findings suggest that interconnected mechanisms might poise K13 mutant parasites to survive DHA treatment and increase the parasite's proteostatic capacity. These mechanisms include an enhanced ability to eliminate damaged proteins through the unfolded protein response and ubiquitin–proteasomal machinery, and remodeling of secretory and vesicular transport processes that impacts hemoglobin endocytosis and protein and lipid trafficking (Fig. 6). K13 mutations may also facilitate rapid recovery post ART exposure by adjusting their central carbon-linked pyruvate and glutamate metabolism, and enabling mitochondrial sensing of cellular damage and the protection and subsequent survival of a subset of drug-exposed parasites (Fig. 6). Further investigation into these mechanisms, which will require additional validation, promises to deliver new approaches to target ART-resistant *P. falciparum*, a goal that is increasingly important given the global dissemination of drug-resistant malaria.

## Methods

**Parasite lines and culture conditions**. *P. falciparum* parasites were cultured at 3% hematocrit in human O$^+$ RBCs (Interstate blood bank, USA) and *P. falciparum* culture media comprising RPMI 1640 (Thermo Fisher Scientific) supplemented with 0.5% (w/v) Albumax II, 50 mg/L hypoxanthine, 0.225% NaHCO$_3$, 25 mM HEPES, and 10 mg/L gentamycin[66]. Parasites were cultured at 37 °C in 5% O$_2$, 5% CO$_2$, and 90% N$_2$. Parasite lines were genotyped by Sanger sequencing *k13* to verify their identities before the start of an experiment. Dd2 and Cam3.II parasites were previously reported[16], with Cam3.II$^{WT}$ earlier referred to as Cam3.II$^{rev}$. We note that Dd2 was adapted to culture in 1980, decades before the introduction of ART derivatives, whereas Cam3.II was adapted in 2010 a decade after ARTs entered widespread use in the region[16,67].

**Sample preparation for microarray-based transcriptomics**. To obtain tightly synchronized parasites, the K13 mutant or WT lines Dd2$^{R539T}$, Dd2$^{C580Y}$, Dd2$^{WT}$, Cam3.II$^{R539T}$, and Cam3.II$^{WT}$ (Fig. 1) were doubly synchronized with 5% D-Sorbitol every IDC cycle for at least three cycles, prior to collecting samples for RNA. We confirmed that parasite cultures were highly synchronous by microscopy, which showed that >95% of all parasites were early rings at the start of each time course experiment. We also computationally assessed transcriptomic profiles by applying Spearman rank correlation calculations for each parasite sample to multiple time points across the IDC of a highly synchronized reference transcriptome (see Supplementary Fig. 1b–e). This analysis yielded mean ± SD correlation coefficients of 0.68 ± 0.03 across samples, as expected for highly synchronized cultures based on our earlier studies[68]. RSAs with the synchronized rings used to initiate our transcriptome sampling yielded survival values that agreed closely with our earlier published data[16].

Samples were collected every 8 h throughout the IDC for the Cam3.II$^{R539T}$ and Cam3.II$^{WT}$ lines, and at 0, 3, 6, 24, 32, and 48 hpi for the Dd2$^{R539T}$, Dd2$^{C580Y}$ and Dd2$^{WT}$ lines. After washing the packed RBC pellets with 1× phosphate-buffered saline (PBS) pH 7.4, a 10× pellet volume of TRIzol was added to the pellet and total RNA was extracted using an acidified phenol-chloroform method[69]. Total RNA was reverse transcribed into cDNA using Superscript II reverse transcriptase and the SMART protocol, with template switching at the cDNA 3′ ends[69]. The cDNA was amino allyl-dUTP labeled using 30 cycles of PCR amplification. Four micrograms of each sample was then labeled with Cy5 fluorescent dyes and mixed with an equal amount of a Cy3 labeled pool comprising mixed asexual blood stages of the reference strain 3D7. Samples were hybridized on a custom *P. falciparum* 70-mer long oligonucleotide microarray chip containing 11,400 probes (GEO Platform: GPL18893) at 65 °C for 20 h[29]. This approach was used to generate transcriptional profiles for 156 samples and was chosen as a cost-effective alternative to RNA-seq. Arrays were washed and scanned using the autogain PMT setting on a Tecan PowerScanner.

**Data analyses for transcriptomics**. Data processing used GenePix Pro software (Molecular devices) and the R Bioconductor LIMMA package version 3.10.3. Only spots with flag >0, and a feature intensity greater than 1.5-fold higher than the background intensity in either Cy5 or Cy3 channel, were included. Background correction used the *Normexp* algorithm and probe ratios were normalized for each array using the Loess method[70]. Ratios of all probes mapping specifically to each gene were averaged to generate a list of gene log$_2$-transformed ratios, representing the log$_2$ ratio of transcript abundance in the sample to transcript abundance in the 3D7 reference pool. For each individual isolate, treatment condition, and replicate IDC, we retained genes present in >66% of the time points, i.e. genes were retained only if they were present in at least 5 out of 7 time points, 4 out of 6 time points, or 8 out of 12 time points. Retained genes also had to be present across the three independent experiments for each parasite line to avoid bias introduced by detection in only one biological experiment. Between 19 and 71 genes were filtered out for each replicate group and these comprised mainly multigene families including *rifins* and *stevors*. Missing data were imputed across all stages and replicates belonging to a given parasite line, using the KNN function applied to a nearest neighbor of 10 genes. This method increased robustness with the imputations while maintaining the specificity for each strain. We used the KNN method to impute missing data, as polynomial fitting or Fast Fourier Transformations did not produce good fits if values were missing at the starting or end points or if sample collection was not regularly spaced over time. After imputing missing data, >4990 (90%) *P. falciparum* genes were analyzed for each parasite strain and were annotated accordingly (PlasmoDB v46).

**Generating a high-resolution Dd2 reference transcriptome model**. Because *P. falciparum* gene expression is a sine-wave function, we generated a high-resolution transcriptome of the reference Dd2 line by interpolating RNA expression of each gene at 30-min time intervals (from a dataset of samples harvested every 2 h). These interpolations used linear regression of polynomial fitting with strict cutoffs of Fisher's exact test $P < 0.001$, adjusted $R^2 > 0.7$, and the Bayesian's Information Criterion (BIC) to find the optimum degree of fit for each gene. Coefficients of the polynomial equation were applied to generate log$_2$ expression ratios at 30-min intervals from 0 to 48 hpi, resulting in 96-time points per gene. In total 4342 genes that could be fitted into a polynomial curve were obtained for the Dd2 reference transcriptome. The predicted 96-time point transcriptomes had very high correlations of 0.94 ± 0.05 (mean ± SD) with the actual 24-time-point transcriptomes sampled across the 24 time points. Applying a filter of max-min expression >2-fold gave us 3500 genes that each had a clear peak of expression at a particular stage in the IDC. This was used to calculate the sample age (in hpi) of each parasite sample time point. Principle component analysis (using the TM4 MeV software version 4.8.1) was performed on each sample to identify underlying variables that contributed to expression differences.

To identify DE genes in the K13 mutant lines across the IDC, we performed Student's *t*-tests with replicates of K13 mutant vs. WT samples or between DHA-treated and DMSO vehicle-treated samples, using the log$_2$ RNA expression levels at each sampling time point for rings, trophozoites, and schizonts. To identify genes that showed a significant difference between the K13 mutant and WT lines in their transcriptional response to DHA exposure, we applied two-factor ANOVA with permutations using subtracted log$_2$ RNA expression ratios of drug-treated vs. DMSO control samples. This was performed in each parasite background across all sampling time points or at the initial 3 and 6 h treatment time points. Kyoto Encyclopedia of Genes and Genomes (KEGG), Gene Ontology (GO), and Malaria Parasite Metabolic Pathway (MPMP) gene sets derived from PlasmoDB were used for functional enrichment analyses of the DE genes by hypergeometric testing, as described[29]. DE genes and pathways in K13 mutants vs. WT lines are listed in Supplementary Data 1 and 2. DE pathways in response to DHA in K13 mutants and WT lines are listed in Supplementary Data 6 and 7.

**Sample preparation for untargeted liquid chromatography-tandem mass spectrometry (LC-MS/MS) proteomics**. Cam3.II$^{R539T}$, Cam3.II$^{C580Y}$, and Cam3.II$^{WT}$ parasites were doubly synchronized using 5% D-Sorbitol for each IDC for at least 2–3 cycles. Early-mid rings or trophozoites of each parasite line were harvested by washing with 1× PBS and then lysed with 0.1% saponin in the presence of a protease inhibitor cocktail (Roche). For each parasite line, we harvested ring and trophozoite stages on two independent occasions, except for Cam3.II$^{C580Y}$ trophozoites that were harvested only once. Parasite extracts were lysed to release proteins and 600 μg of protein of each sample was digested with trypsin and processed as described[71]. Briefly, the saponin-treated parasite pellets were lysed with TEG buffer (50 mM Tris-HCl, 0.5 mM EDTA, 5% β-glycerol phosphate, 1% NP-40, pH 7.4) containing protease and phosphatase inhibitors on ice for 10 min. Samples were centrifuged at 20,000*g* for 3 min, and the supernatant collected. The pellet was again resuspended in TEG buffer without NP-40 and sonicated 3 × 15 s on ice and then centrifuged at 20,000*g* for 3 min. An additional extraction step was performed for ring samples where the pellet was resuspended in the TEG buffer with 2% CHAPS and 0.5% SDS, and sonicated 3 × 15 s on ice and then centrifuged at 20,000*g* for 3 min. The two (trophozoite) or three (ring) supernatant fractions were then combined. Protein concentrations in the final lysates were in the range of 1.2–4.7 mg/mL for rings and 4.3–6.0 mg/mL for trophozoites in a final volume of 0.6 mL. 0.6 mg of protein from each sample lysate was dissolved in 1 mL TEG

buffer (10 mM Tris, 5 mM EDTA, 20 mM β-glycerol phosphate, pH 7.4) and denatured at 37 °C for 30 min in the presence of 10 mM DTT and 0.1% SDS. Iodoacetamide (1 M solution) was then added to a final concentration of 100 mM, adjusted to pH 8.0, and samples were incubated for 1 h at room temperature in the dark. Protein was precipitated on ice for 10 min by addition of 100% trichloroacetic acid (at a ratio of 1:3) to the samples and harvested by centrifugation for 5 min at 2000g followed by two washes with 1 mL TEAB (triethylammonium bicarbonate, pH 8.5). All washing buffer was removed before the protein pellets were sonicated in 300 μL of 50 mM TEAB buffer on ice for 3 × 15 s and digested with trypsin (Promega) at a ratio of 1:20 (w/w) trypsin to protein content, overnight at 37 °C on a rotating platform. Samples were then concentrated to about 0.3–0.5 mL in a Speedvac centrifuge.

For each LC-MS/MS experiment, each parasite line was labeled with a unique isobaric tag (tandem mass tag, TMT 6-plex) and pooled to allow direct comparisons of the proteomic levels of the K13 mutants vs. WT samples. Given the small number of samples, it was not possible to introduce a complete randomization. Instead, we assigned different tags to different samples and runs, as detailed in our data deposited to the ProteomeXchange Consortium (see Data availability section below). Reference pools were not used to internally normalize samples. The experimental design flow chart is shown in Fig. 1. After incubation with TMT reagents, the labeled samples were pooled and concentrated to a volume of 1.2 mL in a Speedvac centrifuge, then mixed with 0.3 mL of 100% acetonitrile and diluted with 10 mM TEAB, pH 8.0 to a final volume of 3 mL. Samples were fractionated using a Resource Q anion-exchange column (GE Healthcare). Peptides were eluted by a linear gradient of 0–1 M NaCl in 10 mM TEAB, pH 8.0, collecting 1 mL fractions. The fractions were concentrated and run on the LTQ-Orbitrap-Velos mass spectrometer (Thermo Scientific). The raw data file was processed using Proteome Discoverer version 2.1 (Thermo Fisher Scientific), searching each file using Mascot version 2.6.0 (Matrix Science Ltd.) against the UniProtKB-Swiss Prot database or a decoy database. To minimize variability, samples were combined on a single MS/MS run. Of note, we identified very few peptides that lacked a TMT tag. Given the abundance of tagged peptides, we estimated the labeling efficiency as >99%.

**Data analysis for proteomics.** Scaffold Q+ (version Scaffold 4.8.2, Proteome Software Inc., Portland, OR) was used to quantify TMT label-based peptides. We applied the following criteria to confidently assign proteins and generate the global protein expression levels. Peptide identifications were accepted if they could be established at >95% probability. Peptide probabilities from X! Tandem were assigned by the Peptide Prophet algorithm[72] with Scaffold delta-mass correction. Protein identifications were accepted if they could be established at >48% probability to achieve a false discovery rate <1.0%, and contained at least one identified peptide unique to the assigned protein. Protein probabilities were assigned by the Protein Prophet algorithm version 5.0 (ref. [72]). Proteins that contained similar peptides and could not be differentiated based on MS/MS analysis alone were grouped to satisfy the principles of parsimony. Normalization of ion intensities was performed iteratively across samples and spectra, using an ANOVA model to account for variability across MS acquisitions and between channels, as described[73]. Spectra data were log-transformed, pruned of those that matched to multiple proteins, and weighted by an adaptive intensity weighting algorithm. We then applied an iterative normalization procedure for each run. Individual quantitative samples were median-normalized within each acquisition run. Intensities for each peptide identification were normalized within the assigned protein. Supplementary Figure 10 shows the distribution of reporter ion intensities across the TMT channels for each of the four LC-MS/MS runs using Cam3.II[R539T], Cam3.II[C580Y], and Cam3.II[WT] K13 ring and trophozoite stage parasites.

For every sample, we first removed poor quality spectra (large spectral count variation) within a protein if the standard deviation in normalized intensities was greater than the mean ± 2 SD (standard deviations) for experiments with 6-plex TMT (since intensity between spectra should be similar for a protein). If TMT = 3, we removed bad spectra where the SD of the $\log_2$ fold change across samples was greater than the median $\log_2$ fold change (assuming that differences between strains were consistent with a small SD). On average, for each run 29,259 high-quality spectra (86% of total) were included in the final quantitation. We obtained the median $\log_2$ normalized spectral intensities of exclusive peptides for each protein. In every biological experiment, DE proteins were identified based on performing a t-test for each individual protein between the K13 mutant and WT samples. The $\log_2$-transformed fold change of the normalized intensities was used to plot the heat maps for each protein. See Supplementary Data 3 for protein expression levels.

**Co-IP and mass spectrometry with monoclonal anti-K13 antibodies.** Antibodies were raised against recombinant K13 protein by injecting mice intraperitoneally with two types of immunogens: the BTB-propeller domain (~40 kDa) or the propeller domain only (~32 kDa), as described[41]. Immunoprecipitation (IP) studies were performed using the direct IP kit (Fisher Scientific) according to the manufacturer's instructions. Briefly, parasites were released from infected RBCs using 0.05% saponin in PBS and were washed twice in PBS. Pelleted parasites were then resuspended in Pierce IP Lysis Buffer supplemented with 1× Halt Protease and Phosphatase Inhibitor Cocktail and 25 U Pierce Universal Nuclease, and lysed on

ice for 10 min with frequent vortexing. Samples were centrifuged at 18,400g for 10 min at 4 °C to pellet cellular debris. Supernatants were collected and protein concentrations therein were determined using the DC protein assay kit (Bio-Rad). IPs were performed used 500 μg of lysate per test sample. A mix of K13 monoclonal antibodies E3 and D9 (2.5 μg each per test sample) was used for IP. Antibody coupling to IP columns, IP, and elution steps were performed according to the manufacturer's instructions. Eluates were analyzed by LC-MS/MS to identify immunoprecipitated proteins.

**Analysis of K13 protein interactions.** We filtered for proteins that were detected with at least three peptide counts in at least three out of six independent co-IP experiments and present in more than 35% samples (5 out of 13 samples). STRING version 11.0 based on seven datasets (textmining, experiments, databases, co-expression, neighborhood, gene fusion, and co-occurrence) was used to generate an interactome of the 21 proteins found to interact with K13. We applied an interaction confidence cutoff of 0.4 and a Markov Clustering with an inflation score of 3. These 21 proteins are listed in Supplementary Data 5.

**Sample preparation for untargeted LC-MS metabolomics.** Testing for Mycoplasma was performed using a MycoAlert PLUS Mycoplasma Detection Kit (Lonza) prior to the start of the sample collection. Mycoplasma-free Cam3.II[C580Y] and Cam3.II[WT] parasites were doubly synchronized by 5% D-sorbitol in each generation for at least two generations. In all, 0–3 hpi early rings of each parasite line were treated for 3 h at 70 nM or 350 nM DHA along with vehicle-treated 0.05% DMSO controls in two to three independent experiments with one to three technical replicates. Twenty-four hpi trophozoites were similarly treated with DHA or DMSO control in a single experiment with six technical replicates for each parasite line and were subsequently magnetically enriched using MACS CS columns on the SuperMACS™ II Separator (Miltenyi Biotec, Inc.) to remove uninfected RBCs. This protocol followed standard conditions previously used to study the metabolomes of Pf parasites exposed to various antimalarial agents[40]. The K13 mutant and WT ring-stage parasites were run in parallel for each metabolomic experiment to allow direct comparisons of their metabolomic states and the effects of DHA. Metabolites were examined from the saponin-lysed parasites and not from the culture media. Metabolites from saponin-lysed parasites were extracted as described[40]. Briefly, parasites were lysed in 1 mL of 90% cold methanol, containing 0.5 μM of the internal standard [$^{13}C_4$, $^{15}N_1$]-Aspartate (Cambridge Isotope) to correct for technical variation arising from sample processing in the data analysis phase. Samples were vortexed and centrifuged to remove cell debris. The clarified supernatants were dried under nitrogen prior to resuspension in HPLC-grade water (Chromasolv; Sigma) for LC-MS analysis. Samples were randomized and 10 μL of extract or processing blank was injected for analysis. Metabolites were analyzed using a previously established reversed phase ion-paired method on a Thermo Exactive Plus Orbitrap LC-MS, scanning from 85 to 1000 m/z (R = 140,000) and operating in negative ESI mode[74,75]. Metabolite separation was achieved on a Synergy Hydro-RP column (100 × 2 mm, 2.5 μm particle size; Phenomenex, Torrance, CA). Solvent A was 97:3 water:methanol with 10 mM tributylamine and 15 mM acetic acid, and solvent B was methanol (all solvents were HPLC grade).

**Data analyses for metabolomics.** Data analysis was performed as described[40]. Briefly, raw data files from the Thermo Exactive Plus orbitrap (.raw) were converted to a format compatible with our analysis software (.mzXML) and spectral data (.mzXML files) were visualized in MAVEN version 8.0.3 (ref. [76]). The labeled [$^{13}C_4$, $^{15}N_1$]-Aspartate internal standard intensity was assessed for technical reproducibility. Metabolites (level 1 annotation) were identified using an in-house database/library generated from pure standard compounds processed on the same LC-MS platform[40]. The criteria for positive identification was based on peak proximity to standard retention time, the observed mass falling within 10 ppm of the expected m/z (calculated from the monoisotopic mass), and the signal/blank ratio (minimum of 10,000 ions). Based upon the above criteria, peaks were further manually inspected and demarcated as good or bad based on peak shape. Peak areas were exported into an R working environment (http://www.R-project.org) to calculate $\log_2$ fold changes for each sample compared to control. Metabolites that were not reliably detected across trials were removed prior to additional analysis to minimize the number of "0" values and subsequent imputation bias. The peak areas for any remaining metabolites not detected ("0" value) were imputed to have 10,000 ions, and metabolites that were negative after blank subtraction (sample area < blank area) were maintained as "0" prior to averaging and $\log_2$ calculation.

To investigate the effect of K13 mutations on P. falciparum physiology, the $\log_2$ fold change of mutant vs. WT metabolites was calculated for each experimental run using the technical replicate peak areas. Three independent experiments, with two to three technical replicates for samples without DHA treatment and one to three replicates for samples after pulsing with DHA, were conducted with ring stages of Cam3.II[C580Y] and Cam3.II[WT] parasites. Spectral data for each technical replicate peak area across all independent trials are listed in Supplementary Data 4. To examine the effects of K13 mutation, we applied metabolomic set enrichment analyses and partial least-squares discriminant analyses (MetaboAnalyst 3.0 package in R). For DHA treatments, the spectral peak areas were averaged and

divided by the average of the DMSO controls for each sample type. Data were log$_2$ transformed to obtain a single log$_2$ fold change for each sample.

**Drug susceptibility assays.** Drug stocks of DHA, atovaquone, fusidic acid, ribavirin, and immucilins G and H were made in dimethyl sulfoxide (DMSO) or ultrapure water (immucilins exclusively), and aliquots were stored at $-20\,°C$. All drug assays were conducted such that the final DMSO concentration was <0.5%. Except for immucilins, all assays used standard media as described above. For immucilin assays, we first adapted parasites to low 10 μM hypoxanthine media, as high hypoxanthine levels have been shown to artificially elevate the IC$_{50}$ values of purine salvage inhibitors. We also used blood washed several times in low hypoxanthine media. For standard IC$_{50}$ dose–response assays, ring-stage cultures at 0.4% parasitemia and 1% hematocrit were exposed for 72 h to a range of 10 drug concentrations that were twofold serially diluted, along with drug-free controls. The IC$_{50}$ defines the drug concentration that results in 50% inhibition of parasite growth. IC$_{50}$ can also be reported in the literature (e.g. ref. [24]) as EC$_{50}$. In the 4 h pulsed drug assays, we obtained 0–3 hpi young rings using Percoll-Sorbitol synchronization[77] and exposed these to the panel of inhibitors. At the end of the pulse, drug was removed by carrying out three washes and a plate transfer using a Tecan EVO 100 automated liquid handler, and parasites were cultured for another 68 h. At least three independent biological replicates were performed, each with technical duplicates. Parasite survival was assessed on an Accuri C6 flow cytometer (BD Biosciences) using SYBR Green I and MitoTracker Deep Red FM (Thermo Fisher Scientific) as stains for DNA and cell viability, respectively. Flow counts were analyzed by gating the live parasites using FlowJo software (FlowJo LLC) as illustrated in Supplementary Fig. 11. Percent parasite survival (normalized to 100%) was plotted against log drug concentrations. A nonlinear regression model was used to determine IC$_{50}$ values (GraphPad Prism). Statistical analyses by one-sample $t$-tests were used to determine whether there was a significant difference in the response of K13 mutant and WT parasites to these individual compounds.

**Drug-pair combination susceptibility assays.** Parasites were exposed for 4 or 72 h to a range of ten twofold serially diluted concentrations of atovaquone plus DHA, applied in fixed molar ratios (1:0, 4:1, 2:1, 1:1, 1:2, 1:4, 0:1) of their individual IC$_{50}$ values. For the isobologram analyses, we calculated the fractional IC$_{50}$ values of the fixed ratios of drug combinations vs. DHA and atovaquone alone. The Combination Index for the *Loewe Additivity* was also used to determine whether the relationship between DHA and the test inhibitor was synergistic, additive, or antagonistic. Paired $t$-tests were used to test for a difference in the combination index values between each mutant and WT line.

**Reporting summary.** Further information on research design is available in the Nature Research Reporting Summary linked to this article.

## Data availability
Microarray gene expression data are available in NCBI's Gene Expression Omnibus with the identifier GSE151189. Data for mass spectrometry proteomics are available in ProteomeXchange Consortium (http://www.proteomexchange.org/) via the PRIDE repository database with the dataset identifier PXD019612 and https://doi.org/10.6019/PXD019612 (https://www.ebi.ac.uk/pride/archive/projects/PXD019612). LC-MS/MS metabolomics data are available at the NIH Common Fund's National Metabolomics Data Repository (NMDR) website, the Metabolomics Workbench (https://www.metabolomicsworkbench.org) with the project ID PR000864, and https://doi.org/10.21228/M80T2X. Datasets extracted from PlasmoDB (https://plasmodb.org), Kyoto Encyclopedia of Genes and Genomes (https://www.genome.jp/kegg/), Gene Ontology (http://geneontology.org/), Malaria Parasite Metabolic (https://mpmp.huji.ac.il/), and STRING (https://string-db.org/) can be accessed using these weblinks. The authors declare that all other data supporting the findings of this study are available within the paper and its Supplementary Information files. Requests for resources and reagents should be directed to D.A.F. (df2260@cumc.columbia.edu). Source data are provided with this paper.

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

## Acknowledgements

We are grateful to Dr. Emily Chen at the CUIMC core facility for performing the mass spectrometry runs for the co-IP experiments and to Dr. Mahmood Alam for advice on proteomic analyses and parasite sampling. We are thankful to Dr. Vern L. Schramm, Albert Einstein College of Medicine, USA, for providing immucilin H and G compounds. This work was supported by the National Institute of Allergy and Infectious Diseases at the National Institutes of Health (R01 AI109023 to D.A.F.), the Department of Defense (Discovery Award W81XWH-19-1-0086 to D.A.F.), and in part by the Intramural Research program of NIAID/NIH. S.M. is a recipient of the Human Frontier Science Program Long-term Postdoctoral Fellowship LT000976/2016-L. B.H.S. received funding from T32 AI106711 (PD: D.A.F) that supports the Columbia University Graduate Training Program in Microbiology and Immunology. We also thank the Huck Institutes of Life Sciences Metabolomics Core Facility at Penn State University.

## Author contributions

S.M., D.A.F., Z.B., and R.M.F. conceived and designed the study. S.M. performed research, acquired, and analyzed data. B.H.S. acquired and analyzed data. N.F.G., L.S.R., T.Y., C.A., E.A., L.S., A.R.B. and J.T. acquired data. S.M., A.B.T., Z.B., M.L., and D.A.F. contributed reagents and to intellectual discussions. S.M. and D.A.F. wrote the manuscript with input from all the authors.

## Competing interests

The authors declare no competing interests.
