## [Peer Review File · Nature Communications]

Reviewer comments, first round -

Reviewer #1 (Remarks to the Author):

NCOMMS

Artemisinin-resistant K13 mutations rewire Plasmodium falciparum's intra-erythrocytic metabolic program to enhance survival

Sachel Mok¹, Barbara H. Stokes¹, Nina F. Gnädig¹, Leila S. Ross¹, Tomas Yeo¹, Chanaki Amaratunga², Erik Allman³, Lev Solyakov⁴, Andrew R. Bottrill⁴, Jaishree Tripathi⁵, Rick M. Fairhurst^{2§}, Manuel Llinás³, Zbynek Bozdech^{5¶}, Andrew B. Tobin^{6¶}, David A. Fidock^{1,7*}

Dear Authors,

This is an interesting study on K13 mutant based effect of antimalarial on *P. falciparum*. This study has potential for understanding the role of k13 in parasite survival and resistance to artemisinin; however, there are several queries and suggestions regarding proteomics and other "omics" datasets.

Here are few comments and suggestions:

(A) PROTEOMICS-study

1. Proteomics study was performed using label-based TMT quantitative proteomics. How was the labeling efficiency? Please include chromatograms of few runs and reporter ion intensity in supplementary.

2. In line 401 – "Missing data was imputed using the KNN function, using a nearest neighbor of 10 genes, to increase robustness of imputation while maintaining the specificity for each strain." Missing value imputation was performed on each group or all together? What was the cut off used (e.g. 70%..), clarify in text?

3. Currently Plasmodb has version 46, the one you have used is v28, were you able to see if there is any upgrade or change in annotations or results you have found with respect to the current release?

4. In para "Preparation of samples for untargeted LC-MS/MS proteomics" TMT 6-plex experimental design were samples randomized to avoid tag biasness? Experimental design included "reference pools" for normalizing each reaction in TMT experiment?

5. In line 459 – "Normalization was performed iteratively (across samples and spectra) on intensities, as described." Brief description regarding the normalization strategy will be useful.

6. In line 462 – "removed poor quality spectra (large spectral count variation) within a protein if the standard deviation in normalized intensities was greater than the mean±2SD (standard deviations)" Spectra for peptides which are showing high variation, are they shared or unique for the protein. As the peptide spectra of shared peptides might vary hugely in case of protein level change in different conditions, removal of which might lead to removal of significant protein from your study.

7. As mentioned regarding the database used for Proteomics study is UniProtKB-Swiss Prot database or a decoy database, were spectral information about K13 mutant variants included, to cross validate the k13 mutant protein and its effect on the protein as such? This may not be linked to the effect of mutants on the rest of the protein profile but will help in understanding about various details like – copy number of k13 and variants in a single organism. Please clarify.

METABOLOMICS study

8. In line 505 – “Metabolites from saponin-lysed parasites were extracted with cold methanol along with spike-in control ...”. You have used parasite metabolites inside the parasite, but there are metabolites which are involved in host protein interactions as well which too contributes to pathobiology. So in the culture media were you able to see any difference or alteration of metabolites for the cases studied?

9. In line 506 –“... along with spike-in control [13C4, 506 15N1] - Aspartate 507 (Cambridge Isotope) as an internal LC/MS standard to correct for technical variation arising from sample processing in the data analysis phase, as described previously⁶⁰”. Is a non-complex internal standard which might not give you a proper idea about the quality of sample processed reliably? Did you use any other reference standards to check the quality of data?

10. Were Metabolite samples run in technical replicates? If yes, please show the chromatograms and CV in supplementary.

11. Which metabolomics database was used for Plasmodium metabolite identification? And how did you annotate the metabolites to know if they were really the same metabolites you got. Annotation is important you can look into Metabolomics Standards Initiative (MSI) and see if your data is well annotated and matching with the claims of your study.

OTHER comments

12. K13 in *P. falciparum* is found to have over 120 mutations, which are linked with Geography as well. It will be useful to highlight that aspect and the predominant type of mutations you found in your studies. Also were you able to check for all and got prominent few or any other approach was taken?

13. It will be useful for relating the importance of the percentage of population survived if exact numbers related to the total population infected and survived can be mentioned along with percentage values. For eg. Line 34-36 - “K13 R539T confers the highest level of resistance across strains (19-49% survival) whereas K13 C580Y.... phenotype (4-24% survival)”.

14. Have you observed copy number alteration of k13 gene, in the strains you isolated/cultured or any literature reporting the same? And possible impact of k13 copies with compensating mutations for pathogen survival.

15. The data of RNA transcriptomics is mean centered which varies highly with outliers. A table or a box plot with outliers for the top 5 or 10 most significant genes, might be helpful to give a better sense on synchronized culture sets. It can be added to supplementary file.

16. In line 376 – “...were doubly synchronized with 5% D-Sorbitol every IDC cycle for at least 3 cycles, prior to collecting samples for RNA.” How did you confirm if the cells are synchronized, which technology and what were the chances to see different stage in the synchronized cell lines? What measure was taken to consider variation in transcript level due to presence of different stages?

17. In supplementary 1a. the replicates have variation in the zero hour values of Fourier Transformed data. What is the reason for the same? As zero hour varies a lot and if not normalized may lead to error replication in rest of the time point readings.

Minor comments:

18. Supplementary 7a. WT vs R539T is more significant as per the graph whereas the * says it otherwise. Also in legend provide the p value corresponding to * or **.

19. It is recommended to use error bars in graphs and should be both side extended (+ Standard

error) for better understanding of the overlaps in different datasets comparison e.g. Supplementary figure 2a.

20. Use of words like “handful” is vague, instead you should provide the number of cases reported. Line 36 – “K13-independent ART resistance has been reported, although this is confined to a handful of sampled isolates or in vitro selections”

21. Give full-explanation of abbreviations at the first appearance.

22. As several experiments have been conducted and multiple datasets was generated, it will be useful for reader if you can provide a descriptive experimental plan for each head.

Reviewer #2 (Remarks to the Author):

Drugs are key in our fight against malaria. The most effective drugs are combinations of artemisinin with a partner drug. In recent years it has been shown that parasites in South-East Asia show some level of resistance to these therapies and that this is mediated by mutations in a parasite gene called Kelch13. Although there are a variety of hypotheses, we do not currently understand how these mutations mediate resistance. By understanding the mechanism, we might be better able to prevent stronger and more widely spread resistance from occurring. The authors seek to understand the mechanism(s) through which mutations in the K13 protein contribute to Artemisinin resistance by omic profiling of K13 mutants.

This manuscript addresses a very important question and it does so using well-designed experiments and appropriate analyses. The paper is very well written, with an excellent, concise introduction to the area under study. The results underscore the current understanding of complex alterations to parasite biology caused by K13 mutations, adding some new pathways to the list. What remains unclear is to what extent these different pathways are relevant for artemisinin resistance or are side effects of the mutation. This work provides an enrichment to our current understanding and will likely aid in interpretation of future experiments.

Minor issues

As I’m sure the authors are fond of hearing, the transcriptomic technology used here is not cutting edge and has certain limitations compared to RNA sequencing, e.g. low dynamic range, low sensitivity to detection of multigene family members. In this particular case, assuming that var, rifin, stevor etc. genes have little to do with art resistance, this is unlikely to be problematic. However, RNA-seq would do a better job.

Of your interactome analysis you say:

“Our interactome analyses of the 21 proteins that were present in at least half of the six co-IP experiments, and absent in four negative resin controls now reveal they are more likely to interact with each other than by random chance”

By this do you mean that the 21 proteins are more connected in the STRING database than expected by chance? Is the CoIP not better evidence? Could you please explain the motivation and methodology behind this analysis more clearly?

You mention several times that you integrated the multi-omics results. This is generally taken to mean using some mathematical approach to combine the results e.g. <https://www.embopress.org/doi/10.15252/msb.20178124>. Here I think that you have just considered the results of the different analyses in coming to a conclusion.

In the discussion you say:

“Our multi-OMICs analysis of isogenic *P. falciparum* lines expressing mutant or WT isoforms of the

ART resistance determinant K13 reveal a striking array of physiological processes in asexual blood stage parasites that are uniquely altered by K13 mutations”.

Does this mean you think that these processes are only altered by K13 mutations? Or do you mean that they are altered by multiple K13 mutations?

Figure 1a – There are some untidy overlaps of elements e.g. 'DHA' with an arrow.

Figure 1b – 'OMICS' should be 'Omics' or 'omics'. Should this be a table instead of a figure? Italicise 'P. falciparum'. Unresolved PMID in the text.

In general 'OMIC' should be 'omic', it is not an acronym.

Is it reasonable to say that the expression of genes in the IDC is highly stage-specific (line 418)? I think you mean that there is a clear peak of expression at a particular stage, which is an important difference.

Reviewer #3 (Remarks to the Author):

Mok et al present an ambitious study (or series of studies in fact) examining various functional impacts of sequence variants of the Plasmodium falciparum K13 kelch-domain protein on parasite biology. The multiplicity of investigations work against a coherent emergent narrative, so that the paper has a descriptive flavour. However an over-arching hypothesis is hinted at in lines 39-45, but this requires sharpening as a means to bring the findings under an overall framework. A number of important observations are reported. Some of these are surprising, or conflict with the findings of other workers and so require a higher burden of evidence. The paper is of great value to the field, but requires attention to some major issues.

Major suggestion - sharpening the hypothesis.

Lines 39-46 suggest "earlier studies" provide evidence that variant K13 leads to reduced endocytosis, less Fe-2+ haem and thus reduced drug activation. (NB: refs 24-32 are all actually very recent from 2015-2020, so "earlier" is not the correct adjective to use here.) Result - a probability greater than 5% but less than 50% of artemisinin-exposed ring-stages to survive a pulse of artemisinin, whether in vivo or in vitro. But these studies all suggest a range of other adaptations (lines 43-45). The authors then state "These processes could offset widespread protein damage elicited by this potent drug." This falls short, just, of stating an interesting question that could help to bind the story together:

"Are the apparent adaptations / enhanced eIF2- phosphorylation / decreased levels of protein ubiquitination / altered rates of proteasome-mediated protein turnover / PI3K-dependent intracellular signaling linked to amplified PI3P vesicles / ... each DIRECTLY caused by variations in K13 sequence (C580Y, R5398T) or are they epistatic or even epigenetic effects enshrined at different loci?"

This question recognises that direct K13 effects could be exerted through the endocytosis / Hb-Fe2+ dampening alone, but that the impact of this lesion is so profound that other preexisting or newly acquired adaptations are required to render the new parasite genotypes competitive in a population. (For evidence that pre-existing adaptations may have a role see <https://pubmed.ncbi.nlm.nih.gov/32755588/>.) Alternatively, the single amino acid K13 lesions which permit partial survival of parasites following ring-stage artemisinin exposure have profound and pleiotropic effects across multiple functional genes. This question would make an effective fulcrum on which to balance the complex set of investigations presented by the authors. Epistasis/epigenetics? Pleiotropy? Or a combination of both? The two explanations do not have to be mutually exclusive.

Throughout the paper the authors imply their view is that pleiotropy of K13 is the dominant

explanation, but in a few cases seem to lean more on epistasis (e.g. lines 282-286). The Dd2 - Cam3.II differences of course support the epistasis/epigenetics view, but in general I think it is hard to distinguish these two possibilities from the untargetted OMICS approach, although this approach is perfectly appropriate to identify the areas for future experimental work. The results presented by Dr Mok and colleagues are of great value to the field and technically challenging across a range of experimental systems - but these deserve to be presented around a stronger philosophical framework.

Major comments on main findings

1) K13 rewires the *P. falciparum* transcriptome.

The authors should consider whether their data rules out the possibility that variant K13-mediated reduction in Hb endocytosis leads to a state of semi-starvation in the early rings, which slows or delays a variety of processes requiring nutrients thus obtained in lesser amounts. Transcriptional responses to falling protein levels could explain many of the differences between transcriptomes in the WT vs K13 variant isogenic pairs. Certainly starvation at ring-stage should also be mentioned as a possible underlying factor in the metabolomics profiles of parasites carrying variant K13.

2) Co-IP results place mitochondrial factors in the K13 interactome

These data, and those of Gnadig et al (ref 37), are poorly concordant with the findings of other studies (refs 24, 25) which do not find evidence of mitochondrial K13. Gnadig and Mok deploy a panel of bespoke MAb to perform the CoIP, whereas other investigators have used either fluorescent tagging of the bait protein (K13) or the Bio-ID approach, which has the advantage of identifying interactors in a live cell, rather than after lysis. This raises a problem for both fluorescent and MAb-mediated pulldowns - how to ensure protein-protein interactions aren't an artefact of cell death and membrane dissolution, which could lead to ingress into cellular compartments not usually occupied?

Further, I could not find an account of the validation of specificity of the MAbs used here - E3 and D9. Has such a validation been provided in the paper to give the reader surety that interactions described are of high confidence? Have alternative K13 pull-down methods been used in the Fidock lab that independently support K13 interactions with mitochondrial factors?

3) K13 mutant parasites exhibit differential responses to atovaquone -DHA combinations

I found this result surprising, and was unsure that appropriate methodology was deployed. In our studies of synergy between other drugs and quinolones including atovaquone (<https://pubmed.ncbi.nlm.nih.gov/32094134/>) we use exposure of 2.5 life cycles' duration to ensure true potency is captured. Atovaquone is a very slow acting drug against blood stages, and the importance of a 4h exposure is unclear. Yes, Suppl Fig 7a does appear to indicate synergy under these conditions, but panel b, with a 72h exposure already provides evidence that this is transient. Further, you do not provide EC50 estimates for these drugs, but only the ratios - it would be interesting to see these.

Further, in our own experiments with Cam3.II and its isogenic siblings we found all three lines to be significantly resistant to doxycycline in vitro. The mechanism is unknown, but is likely to involve apicoplast and perhaps even mitochondrial functions. Is it possible this confounds the analysis - and might Dd2 (also SE Asian in origin) also be doxycycline resistant? This might mean results would be very different in other parasite lines. To sum up - the mitochondrial findings are of interest but require careful verification.

Minor comments and typos

1) Many in the field have ditched "IC50" and replaced with "EC50" as most susceptibility assays measure both cytotoxic and inhibitory drug effects but do not distinguish them. I would suggest use of EC50 throughout.

2) Line 21 - K13 variant parasites have not technically spread rapidly across the whole GMS - there is clear evidence of spread but also de novo emergence of distinct forms (e.g. Myanmar) so some moderation of this sentence would be appreciated.

3) Line 31 - Please change to: "In cultured *P. falciparum*, resistance (or tolerance) is defined as survival in the ... (RSA) above a cut-off of either 1% or 5%."

4) Line 33 - there is not yet a widely agreed orthodoxy in the RSA - some use a 4h DHA exposure, and both flow cytometry and microscopy are used.

5) Line 46 - not just protein, but other biomolecules may be damaged?

- 6) Line 95 - in Suppl Data 2 there are a lot of P values. Do you provide a statement as to why no correction was made for multiple testing? Some statisticians would require this, but I think a statement of justification is sufficient (exploratory work, hypothesis-generating etc etc).
- 7) Fig 3 d - you should mention in the legend that the IP used MAb E3 and D9.
- 8) Line 174 - this is a good experiment, and informative. Panel 4d should have some quantitative data - number of cells of stage X seen over number counted etc. Just showing a single cell is undergraduate stuff.
- 9) Lines 243-246 - please give actual EC50 estimates, as this difference may be significant statistically but biologically unimportant.
- 10) Lines 280-282 - this sentence needs a rework. Suggest: "Our observation that fewer genes were DE (differentially expressed) in K13 WT vs mutant lines supports the hypothesis that a lower level of activated artemisinin accumulates in the cells of the latter."
- 11) Line 282 - "partial survival" at which hpi?
- 12) Lines 284-286 - similarity in the level of reduction in Hb-derived peptides does not mean the process is identical. What happens in dead parasites in both treated lines? They will contribute to the proteome but will have reduced catabolism surely.
- 13) Line 288 - do you need to state this is in untreated parasites?
- 14) Lines 296-314 - how many of these observations can be linked to starvation / nutrient deprivation / reduced nutrition in K13 mutant parasites at the ring stage?
- 15) Line 477 - "... has been described separately." Gnadig et al was published first.
- 16) Line 732 - Add at the end of this sentence in the Figure legend "in the absence of drug treatment".
- 17) Figure 1b - could pfmdr1 and pfcr1 haplotypes be added to this Table?
- 18) Figure 6 and legend. The right to present this model has been earned, but it is a bit vague. K13 mutations are "associated" with phenotypes ... what does that mean? Only arise in parasites already able to display those phenotypes (epistasis) or directly/indirectly causing those phenotypes (pleiotropy)? How, mechanistically, can K13 variants alter central-carbon metabolism? Or can K13 variants only flourish at a population level in parasites that already have these adaptations? Refer to Xiong et al (Cell Reports 2020) or Sutherland et al (FEMS Micro Reviews 2020 almost in press).
- 19) Suppl Figure 7 - I find the importance of these observations difficult to judge. Would it be possible to curtail x and y axes extremities so the traces are larger (relative to size of Figure)?

Colin Sutherland
LSHTM

REVIEWER COMMENTS

Reviewer #1 (Remarks to the Author):

Summary: This is an interesting study on K13 mutant based effect of antimalarial on *P. falciparum*. This study has potential for understanding the role of k13 in parasite survival and resistance to artemisinin; *however, there are several queries and suggestions regarding proteomics and other "omics" datasets.*

Here are few comments and suggestions:

(A) PROTEOMICS-study

1. Proteomics study was performed using label-based TMT quantitative proteomics. How was the labeling efficiency? Please include chromatograms of few runs and reporter ion intensity in supplementary.

Reply: Our revised manuscript now includes a new Supplementary Fig. 4a and c showing the K13 peptides identified from two separate experimental runs, assayed from ring-stage samples prepared from the three isogenic parasite lines Cam3.II^{WT}, Cam3.II^{R539T} and Cam3.II^{C580Y}. We also show in Supplementary Fig. 4b and d representative chromatograms of the relative ion intensity spectra of a K13 peptide for these samples that were isobarically tagged. Samples were combined on a single MS/MS run to minimize variability. In terms of labeling efficiency, we identified very few peptides that lacked a TMT tag. It is not possible to accurately quantify these untagged peptides, in comparison with their TMT-modified equivalent, as their ionization efficiencies differ. Nonetheless, given the abundance of tagged peptides, we estimate the labelling efficiency as >99%.

Our revised manuscript now includes a new Supplementary Fig. 10 that shows the intensity distribution of peptides as a histogram of all four MS/MS runs of the three Cam3.II lines. The default minimum threshold that we selected is 1% of the most intense fragment ion in the MS/MS spectrum. This standard cut-off was used to remove low-intensity values that can pose problems of reproducibility between runs.

2. In line 401 – “Missing data was imputed using the KNN function, using a nearest neighbor of 10 genes, to increase robustness of imputation while maintaining the specificity for each strain.” Missing value imputation was performed on each group or all together? What was the cut off used (e.g. 70%..), clarify in text?

Reply: Our revised text clarifies that the missing data were imputed across all samples for a given parasite line, i.e. on each group. The cut-off was >66% present for each isolate before performing KNN imputation. This is now written in our Materials and methods section as follows (lines 468-476): “For each individual isolate, treatment condition, and replicate IDC, we retained genes present in >66% of the time points... Missing data were imputed across all stages and replicates belonging to a given parasite line, using the KNN function applied to a nearest neighbor of 10 genes. This method increased robustness with the imputations while maintaining the specificity for each strain.”

3. Currently Plasmodb has version 46, the one you have used is v28, were you able to see if there is any upgrade or change in annotations or results you have found with respect to the current release?

Reply: We have now compared versions 28 and 46. While the total number of genes remained the same (5,712) the descriptions were updated for 360 genes. When comparing between PlasmoDB version 28 and 46, only 71 genes have different descriptions (with only 1 gene changing in GO descriptions and ~10 with previous known GO descriptions removed), 7 genes changed from known to hypothetical conserved proteins (mostly surface antigens) and 282 hypothetical proteins have new descriptions (with only 2 genes changing in GO descriptions, 26 with new GO components, 7 with GO new functions and 5 with new GO processes). We have updated these gene descriptions into our revised Supplementary Data 1, 3 and 5, as well as Supplementary Fig. 3c and Fig. 3d. We also now specify the use of v46 on line 480. We found no change in the results but agree it is important to have the most updated descriptions listed.

4. In para “Preparation of samples for untargeted LC-MS/MS proteomics” TMT 6-plex experimental design were samples randomized to avoid tag biasness? Experimental design included “reference pools” for normalizing each reaction in TMT experiment?

Reply: We thank the reviewer for pointing out this concern about tag bias. Given our fairly small number of samples, it was not possible to introduce a complete randomization. Nonetheless, we were mindful to assign different tags to different samples and runs. For example, Cam3.II^{R539T} was tagged with TMT-127, TMT-129, TMT-131 for the run UL02002LS, TMT-128 for UL02049LS and UL02183LS, and TMT-131 in UL02107LS. This information is listed in our proteomic data deposited to the ProteomeXchange Consortium under the dataset identifier PXS019612. We did not include reference pools for each reaction. This information is now included in our revised manuscript as follows (lines 518-521): “Given the small number of samples, it was not possible to introduce a complete randomization. Instead, we assigned different tags to different samples and runs, as detailed in our data deposited to the ProteomeXchange Consortium (see Data availability section below).”

5. In line 459 – “Normalization was performed iteratively (across samples and spectra) on intensities, as described.” Brief description regarding the normalization strategy will be useful.

Reply: The normalization procedure was performed using the Scaffold software. Our revised Materials and methods section now includes a description of the normalization strategy, as follows (lines 536-541): “Normalization of ion intensities was performed iteratively across samples and spectra, using an ANOVA model to account for variability across MS acquisitions and between channels, as described⁷³. Spectra data were log-transformed, pruned of those matched to multiple proteins, and weighted by an adaptive intensity weighting algorithm. We then applied an iterative normalization procedure for each run. Individual quantitative samples were median-normalized within each acquisition run. Intensities for each peptide identification were normalized within the assigned protein.”

6. In line 462 – “removed poor quality spectra (large spectral count variation) within a protein if the standard deviation in normalized intensities was greater than the mean±2SD (standard deviations)” Spectra for peptides which are showing high variation, are they shared or unique for the protein. As the

peptide spectra of shared peptides might vary hugely in case of protein level change in different conditions, removal of which might lead to removal of significant protein from your study.

Reply: We only considered peptides that had >95% probability of correct identity and that were unique to a single protein. We decided to remove peptides that were shared between proteins as it was then not possible to know which protein was showing variable levels. To clarify, we did not remove proteins where the peptide spectra varied substantially between samples, but rather only filtered out the peptide spectra that showed highly variable intensity levels within a sample. This filtering minimized the detection of false positive changes in protein abundance between K13 lines that might be skewed by spurious spectra data. In total, 9-18% of spectra were removed. We note that filtering out spectra with intensities $> \text{mean} \pm 2\text{SD}$ did not change the number of proteins identified that were used for downstream analyses. This is addressed as part of our reply to comment 5 above. We also now include the following text (lines 551-552): “**On average, for each run 29,259 high-quality spectra (86% of total) were included in the final quantitation.**”

7. As mentioned regarding the database used for Proteomics study is UniProtKB-Swiss Prot database or a decoy database, were spectral information about K13 mutant variants included, to cross validate the k13 mutant protein and its effect on the protein as such? This may not be linked to the effect of mutants on the rest of the protein profile but will help in understanding about various details like – copy number of k13 and variants in a single organism. Please clarify.

Reply: Our new Supplementary Fig. 4a and c, cited in response to Reviewer 1 Comment 1 above, show the location of the 11 unique K13 peptides identified across our three isogenic parasite lines, covering 17% of this protein (124 of 726 amino acids). These peptides were identified using the PlasmoDB P. falciparum 3D7 sequences, using a search algorithm that only retained peptides that were identical across samples. Therefore, none of the 11 peptides span the sequences with polymorphism at residues 580 or 539. The protein coverage achieved nonetheless sufficed to examine differences in protein levels between lines. We clarify the K13 coverage as follows (lines 136-138): “**In these samples, K13 was identified with 11 unique peptides, covering 17% of the protein (Supplementary Fig. 4).**”

METABOLOMICS study

8. In line 505 – “*Metabolites from saponin-lysed parasites were extracted with cold methanol along with spike-in control ...*”. *You have used parasite metabolites inside the parasite, but there are metabolites which are involved in host protein interactions as well which too contributes to pathobiology. So in the culture media were you able to see any difference or alteration of metabolites for the cases studied?*

Reply: This is indeed a keen observation by the reviewer and certainly could provide additional insight to the pathobiology of these parasite; however, media samples from these cultures were not processed for metabolomics. We clarify this point as follows (line 595): “**Metabolites were examined only from the saponin-lysed parasites and not from the culture media.**”

9. In line 506 – “*... along with spike-in control [13C4, 506 15N1] - Aspartate 507 (Cambridge Isotope) as an internal LC/MS standard to correct for technical variation arising from sample processing in the data analysis phase, as described previously*”. *Is a non-complex internal standard which might not give you a proper idea about the quality of sample processed reliably? Did you use any other reference standards to check the quality of data?*

Reply: While the use of a single internal standard certainly has caveats, unfortunately it is impossible to correct for or assess all metabolites using uniquely matched labeled reference compounds. Here the use of this single internal standard served as a simple qualitative control, meant to assess technical variation in sample processing and intra-batch analytical drift. This approach is commonly applied in metabolomics and has been previously employed by the authors, as originally cited. Additional quality checks included visual inspection of the peaks, signal/noise ratio, and removal of metabolites that were not reliably detected across 90% of all the trials. This results in high correlation between technical replicates and low variability across experimental trials, as demonstrated in Supplementary Fig. 5b and 5c, respectively.

10. Were Metabolite samples run in technical replicates? If yes, please show the chromatograms and CV in supplementary.

Reply: All of the samples but one were run as technical replicates (n = 1 to 6) as stated in Supplementary Fig. 5a, with n = 1 to 3 biological replicates. Displaying the chromatograms for 63 samples, run across 5 analytical batches and listing the CV for the 96 metabolites detected would be quite complex and not improve the clarity of the manuscript. All metabolomics data displayed in the manuscript denote the measurement listed, the error, and any appropriate statistical methods. Additionally, all data will be made publicly available upon publication, as noted in the manuscript (NIH Metabolomics Workbench (Accession number ST001279)). We have also now added the metabolomics dataset from the database repository in the main manuscript as new Supplementary Data 4, cited on line 146.

11. Which metabolomics database was used for Plasmodium metabolite identification? And how did you annotate the metabolites to know if they were really the same metabolites you got. Annotation is important you can look into Metabolomics Standards Initiative (MSI) and see if your data is well annotated and matching with the claims of your study.

Reply: The metabolites were identified using a previously published, in-house metabolite library derived from running chemical standards on the same analytical platform (Allman et al. 2016 Antimicrob Agents Chemother, PMID 27572391). As a result, matching the features to these pure standards results in level 1 or “identified” compounds, according to the MSI. To highlight that these compounds were annotated from pure standards, we have now edited lines 142-143 to now read: “Targeted metabolomics reveals altered levels of TCA cycle and purine salvage metabolites in untreated Cam3.II^{C580Y} mutants”. Additionally, the reference listed here (#41 in the revised manuscript) is now cited in the metabolomics section to properly refer to this previously published targeted library. This text is now included on lines 144-146: “Metabolomic analyses of Cam3.II^{C580Y} and Cam3.II^{WT} ring (N=3) and trophozoite (N=1) extracts, using a previously published dataset⁴¹, detected a total of 96 metabolites (Supplementary Fig. 5a; Supplementary Data 4).”

Other comments

12. K13 in *P. falciparum* is found to have over 120 mutations, which are linked with Geography as well. It will be useful to highlight that aspect and the predominant type of mutations you found in your studies. Also were you able to check for all and got prominent few or any other approach was taken?

Reply: We appreciate this suggestion and have now added in the following text to highlight that K13 mutations are cluster geographically (lines 29-31): “The degree of resistance is modulated by the specific mutation, which tends to cluster geographically^{7,19}, as well as by the parasite genetic background”. We also added the following text to our Supplementary Fig. 1 legend, as follows: “We note that large-scale genome analysis by the Malaria Genomic Epidemiology Network² has uncovered 92 distinct K13 amino acid substitutions in clinical isolates. These mutations tend to cluster geographically and are almost always present as a single mutation per variant isolate. No copy number variations have been reported. Fewer than ten of these mutations have been confirmed to mediate in vitro resistance or to be associated with delayed parasite clearance³.”

In our study, we selected C580Y as this mutation is the most prevalent in GMS countries such as Cambodia, Vietnam, Thailand, Myanmar, and Laos. We also selected the R539T mutation that is less prevalent but has nonetheless been observed in Cambodia, Vietnam, Myanmar, China and India. We note that in total, only nine of the 92 mutations identified to date have been subjected to gene editing and confirmed to confer increased RSA levels in vitro. At the start of our study, only four had been validated (C580Y, R539T, I543T and Y493H) and we selected the two that either gave the highest degree of resistance (R539T) or was the most prevalent in the field (C580Y). Our lines assayed herein were generated using gene editing, in order to obtain isogenic lines differing only in their K13 sequence, and were sequence-confirmed to harbor only these single point mutations. This is clarified in the revised text as documented below in our response to comment 13.

13. It will be useful for relating the importance of the percentage of population survived if exact numbers related to the total population infected and survived can be mentioned along with percentage values. For eg. Line 34-36 - “K13 R539T confers the highest level of resistance across strains (19-49% survival) whereas K13 C580Y.... phenotype (4-24% survival)”.

Reply: The range of survival percentages presented herein was collated from gene-edited parasites generated in 4-5 strains. We now include information on the number of strains and the geometric mean of the survival assays, as follows (lines 34-38): “K13 R539T confers the highest level of resistance across strains (19-49% survival, with a geometric mean survival of 27%, as determined with four gene-edited strains), and is quite widespread at a relatively low prevalence across Asia²¹. By comparison K13 C580Y, which is the dominant isoform in the GMS, gives a more moderate resistance phenotype (4-24% survival, with a geometric mean survival of 9.2%, as determined with five gene-edited strains)^{15,22}”.

14. Have you observed copy number alteration of k13 gene, in the strains you isolated/cultured or any literature reporting the same? And possible impact of k13 copies with compensating mutations for pathogen survival.

Reply: This is an interesting point given that two studies (Birnbaum et al. 2020 Science; Gnadig et al. 2020 PLoS Pathogens) have demonstrated that expression of a second copy of k13 (either mutant or WT) ablates DHA resistance mediated by endogenous mutant K13. Similarly, a knock-sideways approach that misdirected K13 protein to non-native sites in the parasites also conferred mutant K13-mediated ART resistance, providing evidence for a loss-of-function phenotype (Yang et al. 2019 Cell Reports; Birnbaum et al. 2020 Science). Nonetheless, k13 transcript levels were found to be equivalent between *P. falciparum* artemisinin-resistant and -sensitive isolates in a study of ~1000 clinical samples (Mok et al. 2015 Science). In addition, there have been no reports to date, as best we know, of copy number differences in this gene in clinical isolates or drug-pressured lab lines. As mentioned above in or reply to Comment 12, we now state in the Supplementary Fig. 1 legend: “No copy number variations have been reported”.

15. The data of RNA transcriptomics is mean centered which varies highly with outliers. A table or a box plot with outliers for the top 5 or 10 most significant genes, might be helpful to give a better sense on synchronized culture sets. It can be added to supplementary file.

Reply: We thank the reviewer for raising this point. We mean-centered gene expression data across samples but not across genes within a sample for visualization purposes in Supplementary Fig. 1a. However, our transcriptomic analysis used non mean-centered data to avoid issues with outliers. We have now clarified this in our revised Supplementary Fig. 1 legend, as follows: “To identify DE genes, we used non mean-centered data to avoid issues possibly arising from outliers.” We now also include plots of significant genes in our new Supplementary Fig. 2d, which includes the following legend: “RNA transcript levels for five DE genes that differed significantly between K13 R539T and C580Y mutants relative to WT in at least one sampling time point for Dd2 or Cam3.II parasites. Points and error bars represent the mean \pm SEM of log₂ gene expression levels across three independent experiments for each parasite line; *P<0.05, t-test.” This new panel is cited on line 98 of our revised manuscript.

16. In line 376 – “...were doubly synchronized with 5% D-Sorbitol every IDC cycle for at least 3 cycles, prior to collecting samples for RNA.” How did you confirm if the cells are synchronized, which technology and what were the chances to see different stage in the synchronized cell lines? What measure was taken to consider variation in transcript level due to presence of different stages?

Reply: We confirmed that our parasite cultures were highly synchronous based on microscopy as well as computational assessments of transcriptomic profiles. Microscopic examination of Giemsa smears showed that >95% of all cells were early rings at the start of each time course sampling. Computationally, we determined the level of synchronicity of our parasites by applying Spearman rank correlation calculations for each parasite sample to multiple time points across the intra-erythrocytic developmental cycle of a highly-synchronized reference transcriptome (see Supplementary Fig. 1b-e). We obtained mean \pm SD correlation coefficients of 0.68 \pm 0.03 across samples, as expected for highly-synchronized cultures based on our earlier studies (Foth et al. 2011 Mol Cell Proteomics). In addition, we performed

ring-stage survival assays with the synchronized rings used to initiate our transcriptome sampling, and obtained values that were very consistent with our earlier published data on these lines (Straimer et al. 2015 Science). The revised manuscript now includes this clarification as follows (lines 440-447): “We confirmed that parasite cultures were highly synchronous by microscopy, which showed that >95% of all parasites were early rings at the start of each time course experiment. We also computationally assessed transcriptomic profiles by applying Spearman rank correlation calculations for each parasite sample to multiple time points across the intra-erythrocytic developmental cycle of a highly-synchronized reference transcriptome (see Supplementary Fig. 1b-e). This analysis yielded mean±SD correlation coefficients of 0.68±0.03 across samples, as expected for highly-synchronized cultures based on our earlier studies⁶⁸. RSAs with the synchronized rings used to initiate our transcriptome sampling yielded survival values very consistent with our earlier published data on these lines¹⁵.”

17. In supplementary 1a. the replicates have variation in the zero hour values of Fourier Transformed data. What is the reason for the same? As zero hour varies a lot and if not normalized may lead to error replication in rest of the time point readings.

Reply: Our analysis of the 0h time points collected from different bio-reps and shown for each sample in Supplementary Fig. 1 reveals minimal variance between bio-reps, with a mean±SD correlation of 0.77±0.13 for the Dd2^{WT}, Dd2^{R539T}, Dd2^{C580Y}, Cam3.II^{WT} and Cam3.II^{R539T} lines. By comparison, when compared against the 24h time points, the mean correlation decreased to 0.21. These data speak to the highly-synchronized nature of our cultures, as is also apparent in Fig. 2a. A listing of correlation values for the 0 h time points is provided in our revised Supplementary Fig. 1 legend, as follows: “Our 0h time point data showed minimal variance between bio-reps, with a mean±SD correlation of 0.77±0.13 across lines. By comparison, when compared against the 24h time points, the mean correlation decreased to 0.21.” We also now state (lines 65-68): “Between k13 genotypes, the Dd2 and Cam3.II sets of parasites mapped to highly similar IDC time points, with the isogenic sets of lines differing in developmental age by up to 1.5h and 4h respectively at the 0h start of sampling.”

Minor comments:

18. Supplementary 7a. WT vs R539T is more significant as per the graph whereas the * says it otherwise. Also in legend provide the p value corresponding to * or **.

Reply: Our paired t-test analysis revealed a p value <0.05 when comparing the WT vs. the isogenic R539T line. We now include a revised Supplementary Fig. 9 that better displays these data. Our revised legend now states: “*P<0.05 and **P<0.01 (paired t-test); ns, not significant.”

19. It is recommended to use error bars in graphs and should be both side extended (+ Standard error) for better understanding of the overlaps in different datasets comparison e.g. Supplementary figure. 2a.

Reply: Our revised Fig. 2b and Supplementary Fig. 3b (previously Supplementary Fig. 2a) now show the error bars above and below the means. No error bar could be provided in Supplementary Fig. 3b as this is the absolute number of genes per time point.

20. Use of words like “handful” is vague, instead you should provide the number of cases reported. Line 36 – “K13-independent ART resistance has been reported, although this is confined to a handful of sampled isolates or in vitro selections”

Reply: The four studies cited had very different experimental designs. Demas et al selected for ART resistance, as shown in RSAs (with 7-8% survival), in two Senegalese isolates. Henrici et al. introduced mutations in pfap2-μ and pfubp1 into the 3D7 strain and showed a gain of resistance in the RSA. Mukherjee et al. found three Cambodian isolates lacking K13 mutations that showed low-level ART resistance (RSA survival rates of 1.1%, 3% and 6% (in that study K13 mutations were clearly associated with ART resistance). Rocamora et al. selected very low-level ART resistance, as measured using the RSA, from drug-pressured 3D7 parasites. To clarify this sentence, we have now replaced the former sentence “K13-independent ART resistance has been reported, although this is confined to a handful of sampled isolates or in vitro selections” with (lines 39-41): “K13-independent ART resistance has also

been documented in five field isolates originating from Cambodia or Senegal and in independent selection or transfection studies with the 3D7 parasite line²³⁻²⁶.”

21. Give full-explanation of abbreviations at the first appearance.

Reply: We have gone through the entire manuscript and now assure that all abbreviations are spelled out at time of first use.

22. As several experiments have been conducted and multiple datasets was generated, it will be useful for reader if you can provide a descriptive experimental plan for each head.

Reply: A visual representation of the different omic methods is shown in Fig. 1 and detailed methods are provided for each type of experiment. In light of this Figure and the methods and given the length of the report, we would prefer to not include additional schematics detailing the experimental plan. We have included summaries of each experiment in each of the data submitted to the respective repositories.

Reviewer #2 (Remarks to the Author):

Summary: Drugs are key in our fight against malaria. The most effective drugs are combinations of artemisinin with a partner drug. In recent years it has been shown that parasites in South-East Asia show some level of resistance to these therapies and that this is mediated by mutations in a parasite gene called Kelch13. Although there are a variety of hypotheses, we do not currently understand how these mutations mediate resistance. By understanding the mechanism, we might be better able to prevent stronger and more widely spread resistance from occurring. The authors seek to understand the mechanism(s) through which mutations in the K13 protein contribute to Artemisinin resistance by omic profiling of K13 mutants. This manuscript addresses a very important question and it does so using well-designed experiments and appropriate analyses. The paper is very well written, with an excellent, concise introduction to the area under study. The results underscore the current understanding of complex alterations to parasite biology caused by K13 mutations, adding some new pathways to the list. What remains unclear is to what extent these different pathways are relevant for artemisinin resistance or are side effects of the mutation. This work provides an enrichment to our current understanding and will likely aid in interpretation of future experiments.

Reply: We thank the reviewer for their very positive assessment of our study.

Minor issues

1. As I'm sure the authors are fond of hearing, the transcriptomic technology used here is not cutting edge and has certain limitations compared to RNA sequencing, e.g. low dynamic range, low sensitivity to detection of multigene family members. In this particular case, assuming that var, rifin, stevor etc. genes have little to do with art resistance, this is unlikely to be problematic. However, RNA-seq would do a better job.

Reply: We acknowledge that microarray-based transcriptomics have certain limitations when it comes to detecting low-abundance transcripts, and identifying novel transcripts or alternative splicing events. Nonetheless, the goal of our analyses was to identify genome-wide relative differences in gene expression between K13 mutant and wild-type lines, rather than to quantify absolute transcript levels of genes or to detect multigene families or novel transcripts. Given the very large number of time course samples collected in triplicate across multiple lines, totaling 156 samples, microarray technology presents a much more cost-effective strategy employed herein to effectively answer the question of K13's physiological role. Our revised manuscript now states (lines 458-459): “This approach was used to generate transcriptional profiles for 156 samples and was chosen as a cost-effective alternative to RNA-seq.”

2. Of your interactome analysis you say: “Our interactome analyses of the 21 proteins that were present in at least half of the six co-IP experiments, and absent in four negative resin controls now reveal they are more likely to interact with each other than by random chance”. By this do you mean that the 21 proteins are more connected in the STRING database than expected by chance? Is the CoIP not better evidence? Could you please explain the motivation and methodology behind this analysis more clearly?

Reply: We apologize for the confusing sentence. We observed that the 21 proteins found by K13 co-IPs had more interactions among themselves than expected by random chance from the STRING database (with a protein-protein interaction enrichment p value of 3.14e-7). There were 37 edges found among our list of 21 proteins, which is much higher than that expected for a random set of 21 proteins selected from the entire genome (which would yield an expected 14 edges). The enrichment score therefore suggests that the proteins we detected in our co-IP experiment are at least partially biologically connected as a group. We have now replaced our earlier sentence ~~“Our interactome analyses of the 21 proteins that were present in at least half of the six co-IP experiments, and absent in four negative resin controls now reveal they are more likely to interact with each other than by random chance (STRING; Enrichment $P < 3.1E^{-7}$)”~~ with (lines 169-173): “These experiments yielded 21 proteins that were present in at least half of the six co-IP experiments, and absent in four negative resin controls. Interactome analyses using the STRING database, which uses published biological data to search for observed and predicted protein-protein interactions⁴³, predicted a statistically significant enrichment for interactions among these proteins (protein-protein interaction p value of 3.1e-7)”.

3. You mention several times that you integrated the multi-omics results. This is generally taken to mean using some mathematical approach to combine the results e.g. <https://www.embopress.org/doi/10.15252/msb.20178124>. Here I think that you have just considered the results of the different analyses in coming to a conclusion.

Reply: We thank the reviewer for pointing this out and have edited our text accordingly. Our revised Fig. 1 now states “data analysis” instead of “data integration”. We have also revised the text as (lines 253-254): ~~“Integration of~~ Multi-omics results suggest K13’s involvement in mitochondria-related energetic processes.” This same change is shown in the revised Fig. 5 legend (lines 950-951).

4. In the discussion you say: *“Our multi-OMICs analysis of isogenic P. falciparum lines expressing mutant or WT isoforms of the ART resistance determinant K13 reveal a striking array of physiological processes in asexual blood stage parasites that are uniquely altered by K13 mutations”*. Does this mean you think that these processes are only altered by K13 mutations? Or do you mean that they are altered by multiple K13 mutations?

Reply: We apologize for this lack of clarity in our text. Our Fig. 2d shows processes that were altered in both K13 mutants. We also observed some processes that were altered only in one of the two K13 variants and show those data in Supplementary Data 2. Our revised Fig. 2 legend now states (lines 884-885): “Significantly up-regulated gene sets **common to both** K13 mutants relative to WT parasites.”

5. Figure 1a – There are some untidy *overlaps of elements e.g. ‘DHA’ with an arrow*.

Reply: we thank the reviewer for pointing this out and have now corrected these issues including the DHA text placement in Fig. 1.

6. Figure 1b – *‘OMICs’ should be ‘Omics’ or ‘omics’. Should this be a table instead of a figure? Italicise ‘P. falciparum’. Unresolved PMID in the text.*

Reply: We have now corrected Fig. 1 and converted the former Fig. 1b into a separate Table 1. We have now ensured that the citations and bibliography are correct.

7. *In general ‘OMIC’ should be ‘omic’, it is not an acronym.*

Reply: All instances of ‘OMIC’ in the text have now been changed to ‘omic’.

8. Is it reasonable to say that the expression of genes in the IDC is highly stage-specific (line 418)? I think you mean that there is a clear peak of expression at a particular stage, which is an important difference.

Reply: The reviewer is correct in that we refer to genes having peaks of expression at different times in the IDC. We have replaced the earlier sentence “gave us 3,500 genes ~~that were highly stage-specific at any time point across the 48h IDC~~” with (lines 492-493): “gave us 3,500 genes **that each had a clear peak of expression at a particular stage in the IDC.**”

Reviewer #3 (Remarks to the Author):

Summary: Mok et al present an ambitious study (or series of studies in fact) examining various functional impacts of sequence variants of the Plasmodium falciparum K13 kelch-domain protein on parasite biology. The multiplicity of investigations work against a coherent emergent narrative, so that the paper has a descriptive flavour. However an over-arching hypothesis is hinted at in lines 39-45, but this requires sharpening as a means to bring the findings under an overall framework. A number of important observations are reported. Some of these are surprising, or conflict with the findings of other workers and so require a higher burden of evidence. The paper is of great value to the field, but requires attention to some major issues.

Major suggestion - sharpening the hypothesis.

Lines 39-46 suggest "earlier studies" provide evidence that variant K13 leads to reduced endocytosis, less Fe-2+ haem and thus reduced drug activation. (NB: refs 24-32 are all actually very recent from 2015-2020, so "earlier" is not the correct adjective to use here.) Result - a probability greater than 5% but less than 50% of artemisinin-exposed ring-stages to survive a pulse of artemisinin, whether in vivo or in vitro. But these studies all suggest a range of other adaptations (lines 43-45). The authors then state "These processes could offset widespread protein damage elicited by this potent drug." This falls short, just, of stating an interesting question that could help to bind the story together:

"Are the apparent adaptations / enhanced eIF2- phosphorylation / decreased levels of protein ubiquitination / altered rates of proteasome-mediated protein turnover / PI3K-dependent intracellular signaling linked to amplified PI3P vesicles / ... each DIRECTLY caused by variations in K13 sequence (C580Y, R5398T) or are they epistatic of even epigenetic effects enshrined at different loci?"

This question recognises that direct K13 effects could be exerted through the endocytosis / Hb-Fe2+ dampening alone, but that the impact of this lesion is so profound that other preexisting or newly acquired adaptations are required to render the new parasite genotypes competitive in a population. (For evidence that pre-existing adaptations may have a role see <https://pubmed.ncbi.nlm.nih.gov/32755588/>.) Alternatively, the single amino acid K13 lesions which permit partial survival of parasites following ring-stage artemisinin exposure have profound and pleiotropic effects across multiple functional genes. This question would make an effective fulcrum on which to balance the complex set of investigations presented by the authors. Epistasis/epigenetics? Pleiotropy? Or a combination of both? The two explanations do not have to be mutually exclusive.

Throughout the paper the authors imply their view is that pleiotropy of K13 is the dominant explanation, but in a few cases seem to lean more on epistasis (e.g. lines 282-286). The Dd2 - Cam3.II differences of course support the epistasis/epigenetics view, but in general I think it is hard to distinguish these two possibilities from the untargetted OMICS approach, although this approach is perfectly appropriate to identify the areas for future experimental work. The results presented by Dr Mok and colleagues are of great value to the field and technically challenging across a range of experimental systems - but these deserve to be presented around a stronger philosophical framework.

Reply: We thank the reviewer for these interesting comments. In response, several changes have been made:

i. We have deleted "Earlier" from our sentence that now reads (line 44): "~~Earlier studies~~ Studies have suggested roles for reduced endocytosis..."

ii. Our revised introduction now addresses Reviewer 3's key points about whether K13 has epistatic or pleiotropic relationships to other factors associated with ART resistance and cellular stress responses, as follows (lines 50-52): "One pressing question is whether these cellular effects are a direct result of mutations in K13, or whether they are epistatic phenomena that provide suitable cellular contexts on which mutant K13 can exert ART resistance, or a combination of both."

iii. We further detail the origins of Dd2 and Cam3.II as follows (lines 433-435): “We note that Dd2 was adapted to culture in 1980, decades before the introduction of ART derivatives, whereas Cam3.II was adapted in 2010 a decade after ARTs entered widespread use in the region^{15,38}.”

iv. We also expand on this aspect of pleiotropy vs. epistasis (which as the reviewer astutely observes are non-exclusive) in the Discussion, as follows (lines 308-320): “Our transcriptomic analysis revealed that in the absence of DHA, nearly 400 genes were differentially expressed in Dd2^{R539T} and Dd2^{C580Y} mutant parasites as compared to isogenic Dd2^{WT} parasites, across the 48h IDC. The largest number of DE genes was observed in newly-invaded rings, when K13 mutations impart ART resistance. Among these genes, most were up-regulated in the mutants. Gene set enrichment analysis revealed up-regulation of pathways involved in post-translational modifications (including phosphorylation and palmitoylation), protein export or turnover, lipid metabolism and transport, and the mitochondrial ETC. Many of these pathways were conserved between Dd2 and Cam3.II parasites, although individual genes often differed, suggesting that the genetic background influences the impact of K13 mutations on the parasite transcriptome. These data provide evidence for pleiotropic effects of mutant K13 that are common across parasite strains. Other loci must nonetheless be important in determining the extent to which mutant K13 mediates ART resistance. This is shown by the greater levels of RSA survival in Cam3.II parasites compared to Dd2 (40.2% vs. 19.4% for the two respective R539T mutants¹⁵), suggesting that K13 also has epistatic interactions with other modifier loci.”

v. Our revised Results section also now refers to a DNA damage pathway whose transcript levels differ between Dd2 and Cam3.II as follows (lines 106-109): “Several pathways were also exclusive to K13 mutants in Dd2 but not Cam3.II, including the DNA damage checkpoint (GO process) that contained genes that were up-regulated in ring stages in both Dd2 K13 mutant lines compared to Dd2^{WT}.”

Major comments on main findings

1. K13 rewires the *P. falciparum* transcriptome.

The authors should consider whether their data rules out the possibility that variant K13-mediated reduction in Hb endocytosis leads to a state of semi-starvation in the early rings, which slows or delays a variety of processes requiring nutrients thus obtained in lesser amounts. Transcriptional responses to falling protein levels could explain many of the differences between transcriptomes in the WT vs K13 variant isogenic pairs. Certainly starvation at ring-stage should also be mentioned as a possible underlying factor in the metabolomics profiles of parasites carrying variant K13.

Reply: The reviewer raises an interesting point. We have reviewed an earlier study by Babitt et al. 2012 PNAS (PMID 23112171) that transcriptionally profiled a K13 WT parasite line that was semi-starved by restricting exogenous levels of isoleucine (the only amino acid not present in hemoglobin). Their results indicated a general slowing of transcription across the IDC, without evidence of altered levels of specific pathways. These parasites showed no change in susceptibility to artemisinin. More recently, McLean and Jacobs-Lorena 2017 mBio (PMID 28351924) showed that the Cam3.I^{R539T} Cambodian parasite line (earlier published by our lab) showed a three-fold better rate of recovery following 72h of isoleucine starvation as compared with the isogenic Cam3.I^{WT} Cambodian parasite line. No RSA analysis was conducted. These data suggest a possible impact of mutant K13 on tolerance to amino acid partial starvation, which might relate to the ability of mutant k13 to reduce hemoglobin endocytosis. Our revised Discussion now states (lines 354-361): “One question that merits further exploration is whether the reported impact of K13 mutations on reducing hemoglobin endocytosis^{27,28} might trigger a semi-starvation state that accounts for some of our observed metabolic changes. A recent study showed that the Cam3.I^{R539T} Cambodian parasite line¹⁵ showed a three-fold better rate of recovery following 72h of isoleucine starvation as compared with the isogenic Cam3.I^{WT} Cambodian parasite line, suggesting that K13-mutant parasites might be better able to tolerate semi-starvation⁵⁸. Nonetheless, transcriptomic profiling of amino acid-starved 3D7 WT K13 parasites showed a general slowing of transcription across the IDC⁵⁹, without evidence of specific pathways observed in our study.”

2. Co-IP results place mitochondrial factors in the K13 interactome

These data, and those of Gnadig et al (ref 37), are poorly concordant with the findings of other studies (refs 24, 25) which do not find evidence of mitochondrial K13. Gnadig and Mok deploy a panel of bespoke MAb to perform the CoIP, whereas other investigators have used either fluorescent tagging of the bait protein (K13) or the Bio-ID approach, which has the advantage of identifying interactors in a live cell, rather than after lysis. This raises a problem for both fluorescent and MAb-mediated pull-downs - how to ensure protein-protein interactions aren't an artefact of cell death and membrane dissolution, which could lead to ingress into cellular compartments not usually occupied? Further, I could not find an account of the validation of specificity of the MAbs used here - E3 and D9. Has such a validation been provided in the paper to give the reader surety that interactions described are of high confidence? Have alternative K13 pull-down methods been used in the Fidock lab that independently support K13 interactions with mitochondrial factors?

Reply: Our publication by Gnadig et al. 2020 PLoS Pathogens demonstrated specificity of our anti-K13 monoclonal antibodies by showing concordance between the banding patterns of our antibodies and the bands detected with antibodies to GFP or a 3×HA epitope tag, as demonstrated by Western blot hybridizations of parasites expressing GFP-K13 or 3×HA-K13. Furthermore, Pearson Coefficient Correlation analysis with E3 and D9 monoclonal antibodies showed excellent correlations between the subcellular localization of anti-K13 stained foci and the anti-GFP or anti-3×HA antibody labeled foci. Moreover, the K13 protein was the most highly abundant protein across all independent experiments. We also included negative controls of running the lysates derived from the K13 lines with the control resin in the absence of the K13 antibody and then removed proteins that were present in these pull-downs. We have not used other independent pull-down methods to support K13 interactions with mitochondria factors, as we have been unable to locate any antibodies specific to *P. falciparum* mitochondrial proteins. We agree with the minimal concordance with the Birnbaum et al. and Yang et al. papers cited above as references 24 and 25. We note nonetheless concordance with GFP-K13 co-IP data reported by Siddiqui et al. 2020 mBio (PMID 32098812). We summarize these points in our revised manuscript as follows (lines 167-168): “The specificity of these antibodies was validated using Western blot and IFA data comparing our K13 signals with results from parasite lines expressing GFP-K13 or 3×HA tagged K13⁴².”

We also now state (lines 375-380): “We note that our co-IP results differ from the set of K13-interacting proteins identified using a quantitative dimerization-induced bio-ID approach with GFP-tagged K13²⁷. Partial overlap was nonetheless observed between our study and a separate report that used GFP-Trap beads to affinity purify GFP-K13 and then identified the immunoprecipitated proteins¹⁷. These results merit further investigation using independent approaches such as co-IP data with antibodies specific to mitochondrial proteins, which we are to date unable to obtain.”

3. K13 mutant parasites exhibit differential responses to atovaquone -DHA combinations

I found this result surprising, and was unsure that appropriate methodology was deployed. In our studies of synergy between other drugs and quinolones including atovaquone (<https://pubmed.ncbi.nlm.nih.gov/32094134/>) we use exposure of 2.5 life cycles' duration to ensure true potency is captured. Atovaquone is a very slow acting drug against blood stages, and the importance of a 4h exposure is unclear. Yes, Suppl Fig 7a does appear to indicate synergy under these conditions, but panel b, with a 72h exposure already provides evidence that this is transient. Further, you do not provide EC50 estimates for these drugs, but only the ratios - it would be interesting to see these.

Reply: Our 72h continuous exposure assays provide evidence that atovaquone paired with DHA is slightly more potent against the K13 mutants compared to the WT lines, however the 4h data are clearly more striking in terms of synergy between ATQ and DHA. We tested the 4h exposure to see whether we could detect an early signal following ETC perturbation without killing the parasite as the effect of DHA in the longer assays would be the main effector. Our revised manuscript now reads (lines 261-262): “In contrast, with the slower-acting mitochondrial ETC inhibitor atovaquone^{45,46} (ATQ)...”. Our new

Supplementary Data 8 now provides the IC₅₀ and IC₉₀ values for these drugs tested at 4h and 72h, as stated on line 264: “Individual IC₅₀ and IC₉₀ data are provided in Supplementary Data 8.”

4. Further, in our own experiments with Cam3.II and its isogenic siblings we found all three lines to be significantly resistant to doxycycline in vitro. The mechanism is unknown, but is likely to involve apicoplast and perhaps even mitochondrial functions. Is it possible this confounds the analysis - and might Dd2 (also SE Asian in origin) also be doxycycline resistant? This might mean results would be very different in other parasite lines. To sum up - the mitochondrial findings are of interest but require careful verification.

Reply: We have not tested our parasite lines for doxycycline, as earlier studies from Yeh and DeRisi 2011 PLoS Biol (PMID 21912516) and Uddin et al. 2018 Antimicrob Agents Chemother (PMID 29109165) have both provided evidence of its inhibition of the parasite apicoplast. We are unaware of data implicating its possible inhibition of mitochondria. Our data did not uncover associations with the apicoplast, leading us to focus instead on the several lines of evidence implicating mitochondrial processes. In future studies, we will further explore whether Cam3.II and other contemporary K13-mutant Asian isolates show doxycycline resistance. In reference to the Reviewer's last sentence, we have revised our concluding Discussion sentence from “~~Further elucidation of these mechanisms~~ promises to deliver new approaches to target ART-resistant *P. falciparum*” to (lines 421-422): “Further investigation into these mechanisms, which will require additional validation, promises to deliver new approaches to target ART-resistant *P. falciparum*”. Our revised manuscript also now includes a reference to an article very recently published in Nature Communications that implicates altered mitochondrial physiology in conferring decreased ART susceptibility in *Toxoplasma gondii*, as follows (lines 406-408): “Interestingly, a genome-wide CRISPR/Cas9 screen in the Apicomplexan parasite *Toxoplasma gondii* recently identified components of the TCA cycle and heme biosynthesis as mitochondrial determinants of DHA susceptibility⁶⁶.”

Minor comments and typos

5. Many in the field have ditched "IC50" and replaced with "EC50" as most susceptibility assays measure both cytotoxic and inhibitory drug effects but do not distinguish them. I would suggest use of EC50 throughout.

Reply: We appreciate this point, but would rather maintain the IC₅₀ nomenclature as it is consistent with all our previous studies and it would be unclear why we suddenly changed when the assays used are the same. We now indicate in the methods that IC₅₀ has also been shown as EC₅₀ by some other groups and cite Henrici et al. 2019 Antimicrob Agents Chemother (PMID 31636063) as one example. We now state (lines 629-631): “The IC₅₀ defines the drug concentration that results in 50% inhibition of parasite growth. IC₅₀ can also be reported in the literature (e.g.²⁴) as EC₅₀.”

6. Line 21 - K13 variant parasites have not technically spread rapidly across the whole GMS - there is clear evidence of spread but also de novo emergence of distinct forms (e.g. Myanmar) so some moderation of this sentence would be appreciated.

Reply: We thank the reviewer for this important clarification. Our revised sentence now reads (lines 20-21): “*P. falciparum* resistance to ART, which first emerged in western Cambodia, is now present across the Greater Mekong Sub-region”.

7. Line 31 - Please change to: "In cultured *P. falciparum*, resistance (or tolerance) is defined as survival in the ... (RSA) above a cut-off of either 1% or 5%."

Reply: We are happy to change the sentence to include “or tolerance”. We are hesitant however to list “either 1% or 5%”. The benchmark of 1% has been used in multiple studies, and we prefer to not confuse the reader by including a second, less common threshold. In our assays, performed now for many years, we see >1% survival of assays initiated with highly-synchronized very early ring-stage parasites as being a reasonable benchmark. Our revised sentence now reads (lines 31-32): “With cultured parasites, resistance (or tolerance) is typically defined as >1% parasite survival”

8. Line 33 - there is not yet a widely agreed orthodoxy in the RSA - some use a 4h DHA exposure, and both flow cytometry and microscopy are used.

Reply: We agree and now address this in our revised text as follows (lines 32-33): “the Ring-stage Survival Assay (RSA), in which young rings (0-3h post-invasion (hpi)) are exposed to 700 nM DHA for 4h to 6h”.

9. Line 46 - not just protein, but other biomolecules may be damaged?

Reply: We agree, and now replace the former sentence “These processes could offset widespread ~~protein damage elicited by this potent drug~~” with (lines 49-50): “These processes could offset widespread ART-mediated damage to parasite proteins and other biomolecules^{35,36}.”

10. Line 95 - in Suppl Data 2 there are a lot of P values. Do you provide a statement as to why no correction was made for multiple testing? Some statisticians would require this, but I think a statement of justification is sufficient (exploratory work, hypothesis-generating etc etc).

Reply: We thank the reviewer for this suggestion. Our revised header in Supplementary Data 2 now specifies: “No correction was made for multiple testing, as the goal of this exploratory work was to generate testable hypotheses.”

11. Fig 3 d - you should mention in the legend that the IP used MAb E3 and D9.

Reply: This is now indicated as follows (lines 909-910): “Interactome network of the 21 putative K13 interacting partners detected by co-immunoprecipitation (co-IP) with the K13-specific monoclonal antibodies E3 and D9⁴².”

12. Line 174 - this is a good experiment, and informative. Panel 4d should have some quantitative data - number of cells of stage X seen over number counted etc. Just showing a single cell is undergraduate stuff.

Reply: Our revised manuscript now includes the new Supplementary Fig. 6 (cited on line 208) that shows additional representative examples of parasite morphology following DHA or DMSO treatment for the different parasite lines and time points. We are hesitant to quantify developmental stages at each time point as one cannot always readily discriminate between live and dead parasites such as trophozoites. Our time point-specific transcriptomic analysis is more informative about the IDC cycle in these treated parasites and is present in Supplementary Fig. 1. We have also revised Fig. 4 legend, which now reads (lines 932-933): “These images are representative of the majority of parasites observed at each time and condition per parasite line, with additional examples provided in Supplementary Fig. 6.”

13. Lines 243-246 - please give actual EC50 estimates, as this difference may be significant statistically but biologically unimportant.

Reply: Our revised text now states (lines 264 and 961): “Individual IC₅₀ and IC₉₀ data are provided in Supplementary Data 8”.

14. Lines 280-282 - this sentence needs a rework. Suggest: “Our observation that fewer genes were DE (differentially expressed) in K13 WT vs mutant lines supports the hypothesis that a lower level of activated artemisinin accumulates in the cells of the latter.”

Reply: We thank the reviewer for pointing out this poorly constructed sentence and for suggesting an alternative. Our revised sentence now reads (lines 298-300): “Our observation that fewer genes were DE (differentially expressed) in DHA-treated K13 mutant parasites, as compared to DHA-treated isogenic WT lines, supports the hypothesis that a lower level of activated ART accumulates and exerts its effect in K13 mutant parasites”.

15. Line 282 - “partial survival” at which hpi?

Reply: We have now clarified this sentence to read (lines 300-302): “However, this did not account for the ~~partial survival~~ increased rates of survival of DHA-treated rings (0-6 hpi) elicited by K13 mutations...”

16. Lines 284-286 - similarity in the level of reduction in Hb-derived peptides does not mean the process is identical. What happens in dead parasites in both treated lines? They will contribute to the proteome but will have reduced catabolism surely.

Reply: We agree with the reviewer about the complexities of how parasite death relates to the parasite proteome and catabolism. To avoid this confounder, we had designed our experiment to measure Hb-derived peptide levels after only 3h of DHA treatment (at 70 nM and 350 nM; see Fig. 4a). This is the standard treatment protocol used by the Llinas lab when examining the effect of antimalarials on the parasite metabolome (Allman et al. 2016 Antimicrob Agents Chemother, PMID 27572391). Our revised sentence clarifies this as follows (lines 303-306): "Furthermore, in our study the reduction in hemoglobin-derived peptides upon treatment with DHA, observed within 3h of drug exposure, occurred to a similar degree in both K13 mutant and WT parasites. These results suggest that the mechanism of ART resistance afforded by mutant K13 extends beyond a role for reduced hemoglobin endocytosis in rings."

Also we added (lines 588-590): "This protocol followed standard conditions previously used to study the metabolomes of Pf parasites exposed to various antimalarial agents⁴¹."

17. Line 288 - do you need to state this is in untreated parasites?

Reply: We now make this helpful clarification as follows (lines 328-329): "K13 C580Y and R539T mutations showed very similar impacts on both parasite transcription and translation in untreated parasites."

18. Lines 296-314 - how many of these observations can be linked to starvation / nutrient deprivation / reduced nutrition in K13 mutant parasites at the ring stage?

Reply: Please see our reply to Comment 1 of Reviewer 3 that addresses transcriptomic profiles. We are unaware of any proteomic profiling of nutrient deprived Pf parasites. For metabolomics, the comparison of our data with that published by Babbitt et al (cited above) shows little correlation. We now refer to this as follows (lines 361-363): "Metabolomics data in that study also showed minimal overlap with our results, suggesting that mutant K13-mediated reduction in Hb endocytosis does not readily mimic starvation responses resulting from amino acid deprivation."

19. Line 477 - "... has been described separately." Gnadig et al was published first.

Reply: We thank the reviewer for detecting this outdated statement. Our revised sentence now reads (lines 560-561): "..., as ~~will be~~ described separately⁴²."

20. Line 732 - Add at the end of this sentence in the Figure legend "in the absence of drug treatment".

Reply: This is now added, as follows (lines 857-859): "RNA, protein or metabolite samples were collected for each parasite line at 6-7 time points throughout the 48h asexual blood stage cycle in the absence of drug treatment."

21. Figure 1b - could pfmdr1 and pfcr1 haplotypes be added to this Table?

Reply: These haplotypes have now been added to the footnotes in our new Table 1, which replaces its earlier representation as Fig. 1b.

22. Figure 6 and legend. The right to present this model has been earned, but it is a bit vague. K13 mutations are "associated" with phenotypes ... what does that mean? Only arise in parasite already able to display those phenotypes (epistasis) or directly/indirectly causing those phenotypes (pleiotropy)? How, mechanistically, can K13 variants alter central-carbon metabolism? Or can K13 variants only flourish at a population level in parasites that already have these adaptations? Refer to Xiong et al (Cell Reports 2020) or Sutherland et al (FEMS Micro Reviews 2020 almost in press).

Reply: These are all good points. There is no doubt that K13 mutations can exert widely variable levels of resistance depending on the parasite strain, as we observed in our article by Stramer et al. 2015 Science (PMID 25502314) and evident herein as RSA values differing quite substantially between Dd2

and Cam 3.II (e.g. the R539T mutation provides 19.4% and 40.2% survival in the RSA in these strains, respectively). Our response to these points and the corresponding changes made to the manuscript are detailed above in our response to the Major Suggestion from Reviewer 3. In addition, our revised Discussion now states (lines 321-326): “Intriguingly, K13-mutant Asian parasites were recently found to carry mutant alleles of DNA repair genes that provided enhanced protection against artesunate-mediated DNA damage; several of these genes are also present in Dd2⁴⁹. Other studies have also reported founder genetic backgrounds on which mutant K13 evolved in Southeast Asia, supporting a role for epistatic interactions that contribute to the resistance phenotype and that may help with a process referred to as cellular healing in ART-treated K13-mutant parasites^{16,50,51}.”

23. Suppl Figure 7 - I find the importance of these observations difficult to judge. Would it be possible to curtail x and y axes extremities so the traces are larger (relative to size of Figure)?

Reply: We have curtailed these axes to increase the size of the plots, while keeping enough space to have the insets adequately displayed. We have also revised this figure (renamed as Supplementary Fig. 9) with standardized axes and added a $y=x$ dotted line at 0.5 for improved clarity.

Reviewer comments, second round -

Reviewer #1 (Remarks to the Author):

Authors have substantially revised this manuscript.

However, I still have two concerns:

1. It will be good to mention the limitations of the proteomics investigations of this study in concluding remarks, which includes number of replicates, labelling efficiency tests, lack of reference pools etc.
2. Metabolomics technical replicates are also not available for all the analysis, which should be highlighted.

Reviewer #3 (Remarks to the Author):

The authors have provided a pleasing response to my critical review of the first submission, and I am satisfied no major issues remain. The second paragraph of the Discussion constitutes a helpful discussion of the pleiotropy/epistasis dichotomy in K13-mutant artemisinin resistance, which I think readers will find of interest.

I also note a thorough and enlightening response to the critique of Reviewer 1, particularly around details of the proteomic data.

A mere handful of minor corrections should be addressed before publication:

1. Line 283 - I remain unsure of the biological importance of the atovaquone data, interesting though it is. The phrase "... might prevent the recovery of..." could be moderated to "... can reduce" or similar.
2. Line 298 - you have the cart before the horse: please swap "DE" and "differentially expressed" so that "DE" appears in the brackets.
3. Lines 406-408. I am aware of this work in *Toxoplasma* - and my first thought was that care should be taken in interpreting this as a model for artemisinin action in *Plasmodium*. Lack of haemoglobin ingestion and metabolism greatly reduces intracellular opportunities for endoperoxide activation in the former genus. Can such a caveat be added to your argument here?
4. Lines 419 - 421. The sentence here was clearly meant to be deleted, as it precedes its own replacement.

Colin Sutherland

Response to - REVIEWERS' COMMENTS

Reviewer #1 (Remarks to the Author): *Authors have substantially revised this manuscript. However, I still have two concerns:*

1. *It will be good to mention the limitations of the proteomics investigations of this study in concluding remarks, which includes number of replicates, labelling efficiency tests, lack of reference pools etc.*

Reply: In our Methods section, we now include the following sentence about replicates (lines 514-6): “For each parasite line, we harvested ring and trophozoite stages on two independent occasions, except for Cam3.II^{C580Y} trophozoites that were harvested only once.” For the labeling efficiency, we now state (lines 552-3): “Of note, we identified very few peptides that lacked a TMT tag. Given the abundance of tagged peptides, we estimated the labelling efficiency as >99%.” Regarding reference pools, we now state (line 543): “Reference pools were not used to internally normalize samples.” In our Discussion, we now state (lines 333-6): “In considering these data, we note that experiments had a limited number of independent repeats (generally two) and did not include reference pools or labeling efficiency tests. High reproducibility was nonetheless observed between repeats (Fig. 3a and Supplementary Fig. 3b).”

2. *Metabolomics technical replicates are also not available for all the analysis, which should be highlighted.*

Reply: Our revised Methods section now states (lines 652-5): “Three independent experiments, with two to three technical replicates for samples without DHA treatment and one to three replicates for samples after pulsing with DHA, were conducted with ring stages of Cam3.II^{C580Y} and Cam3.II^{WT} parasites. Spectral data for each technical replicate peak area across all independent trials are listed in Supplementary Data 4.”

Reviewer #3 (Remarks to the Author):

The authors have provided a pleasing response to my critical review of the first submission, and I am satisfied no major issues remain. The second paragraph of the Discussion constitutes a helpful discussion of the pleiotropy/epistasis dichotomy in K13-mutant artemisinin resistance, which I think readers will find of interest. I also note a thorough and enlightening response to the critique of Reviewer 1, particularly around details of the proteomic data. A mere handful of minor corrections should be addressed before publication:

1. Line 283 - *I remain unsure of the biological importance of the atovaquone data, interesting though it is. The phrase "... might prevent the recovery of..." could be moderated to "... can reduce" or similar.*

Reply: We agree and have now reworded our text as follows (lines 281-3): " These results suggest that ATQ-mediated inhibition of the mitochondrial cytochrome *bc*₁ Q_o site ~~might prevent~~ **can reduce** the recovery of K13 mutant ring-stage parasites following treatment with pro-oxidant ART drugs."

2. Line 298 - *you have the cart before the horse: please swap "DE" and "differentially expressed" so that "DE" appears in the brackets.*

Reply: We thank the reviewer for pointing this out and have corrected this (see line 297).

3. Lines 406-408. *I am aware of this work in Toxoplasma - and my first thought was that care should be taken in interpreting this as a model for artemisinin action in Plasmodium. Lack of haemoglobin ingestion and metabolism greatly reduces intracellular opportunities for endoperoxide activation in the former genus. Can such a caveat be added to your argument here?*

Reply: We agree that unlike in *Plasmodium falciparum*, *Toxoplasma gondii* does not have features of hemoglobin digestion, a process that releases free heme required to activate artemisinin drug. For clarity, we now state (lines 410-2): "**One important distinction, however, is that unlike *Plasmodium*, *Toxoplasma* parasites do not endocytose or metabolize hemoglobin, thereby removing this path for ART activation.**"

4. Lines 419 - 421. *The sentence here was clearly meant to be deleted, as it precedes its own replacement.*

Reply: We apologize for the duplication and have now removed this sentence from the text.